# Bioactive-Loaded Hydrogels Based on Bacterial Nanocellulose, Chitosan, and Poloxamer for Rebalancing Vaginal Microbiota

**DOI:** 10.3390/ph16121671

**Published:** 2023-11-30

**Authors:** Angela Moraru, Ștefan-Ovidiu Dima, Naomi Tritean, Elena-Iulia Oprița, Ana-Maria Prelipcean, Bogdan Trică, Anca Oancea, Ionuț Moraru, Diana Constantinescu-Aruxandei, Florin Oancea

**Affiliations:** 1Faculty of Biotechnologies, University of Agronomic Sciences and Veterinary Medicine Bucharest, Bd. Mărăști Nr. 59, Sector 1, 011464 Bucharest, Romania; angela.moraru@pro-natura.ro; 2S.C. Laboratoarele Medica Srl, Strada Frasinului Nr. 11, 075100 Otopeni, Romania; ionut.moraru@pro-natura.ro; 3Polymers and Bioresources Departments, National Institute for Research and Development in Chemistry and Petrochemistry—ICECHIM, Splaiul Independentei Nr. 202, Sector 6, 060021 Bucharest, Romania; ovidiu.dima@icechim.ro (Ș.-O.D.); naomi.tritean@icechim.ro (N.T.); bogdan.trica@icechim.ro (B.T.); 4Faculty of Biology, University of Bucharest, Splaiul Independentei Nr. 91-95, Sector 5, 050095 Bucharest, Romania; 5Department of Cellular and Molecular Biology, National Institute of Research and Development for Biological Sciences, Splaiul Independentei Nr. 296, Sector 6, 060031 Bucharest, Romania; iulia.oprita@incdsb.ro (E.-I.O.); anamaria.prelipcean@incdsb.ro (A.-M.P.); oancea.anca@gmail.com (A.O.)

**Keywords:** mucoadhesion, biocompatible, prebiotics, probiotic growth, antimicrobial, thyme essential oil, *Vitis vinifera*, *Opuntia ficus-indica*, lactic acid, citric acid

## Abstract

Biocompatible drug-delivery systems for soft tissue applications are of high interest for the medical and pharmaceutical fields. The subject of this research is the development of hydrogels loaded with bioactive compounds (inulin, thyme essential oil, hydro-glycero-alcoholic extract of *Vitis vinifera*, *Opuntia ficus-indica* powder, lactic acid, citric acid) in order to support the vaginal microbiota homeostasis. The nanofibrillar phyto-hydrogel systems developed using the biocompatible polymers chitosan (CS), never-dried bacterial nanocellulose (NDBNC), and Poloxamer 407 (PX) incorporated the water-soluble bioactive components in the NDBNC hydrophilic fraction and the hydrophobic components in the hydrophobic core of the PX fraction. Two NDBNC-PX hydrogels and one NDBNC-PX-CS hydrogel were structurally and physical-chemically characterized using Fourier-transform infrared (FTIR) spectroscopy, X-ray diffraction (XRD), transmission electron microscopy (TEM), and rheology. The hydrogels were also evaluated in terms of thermo-responsive properties, mucoadhesion, biocompatibility, and prebiotic and antimicrobial effects. The mucin binding efficiency of hydrogel base systems was determined by the periodic acid/Schiff base (PAS) assay. Biocompatibility of hydrogel systems was determined by the MTT test using mouse fibroblasts. The prebiotic activity was determined using the probiotic strains *Limosilactobacillus reuteri* and *Lactiplantibacillus plantarum* subsp. *plantarum.* Antimicrobial activity was also assessed using relevant microbial strains, respectively, *E. coli* and *C. albicans*. TEM evidenced PX micelles of around 20 nm on NDBNC nanofibrils. The FTIR and XRD analyses revealed that the binary hydrogels are dominated by PX signals, and that the ternary hydrogel is dominated by CS, with additional particular fingerprints for the biocompounds and the hydrogel interaction with mucin. Rheology evidenced the gel transition temperatures of 18–22 °C for the binary hydrogels with thixotropic behavior and, respectively, no gel transition, with rheopectic behavior for the ternary hydrogel. The adhesion energies of the binary and ternary hydrogels were evaluated to be around 1.2 J/m^2^ and 9.1 J/m^2^, respectively. The hydrogels exhibited a high degree of biocompatibility, with the potential to support cell proliferation and also to promote the growth of lactobacilli. The hydrogel systems also presented significant antimicrobial and antibiofilm activity.

## 1. Introduction

Biocompatible hydrogels are intensively studied due to their wide potential in medical and pharmaceutical fields as drug delivery systems via various pathways and mechanisms, from oral, nasal, intravenous, or injectable targeted delivery, to skin and soft-tissue topical delivery [1,2]. Therefore, hydrogel formulations can be specifically designed for particular applications and can take various shapes, such as nano/micro vesicles or capsules, films and patches, gels, cryogels, microneedles, and 3D and 4D shapes [3,4,5,6,7,8,9,10,11], while preserving some general characteristics such as biocompatibility, bioactivity, and stability. In particular, hydrogels for soft tissue and transdermal delivery of bioactive formulations require mucoadhesive properties in order to increase the transfer surface and contact time, together with additional requirements of pH-dependent stability and mechanical resistance [12,13,14,15,16,17].

A number of biopolymers and biocompatible synthetic polymers are frequently used in various thermo- and pH-responsive bio(nano)formulations, such as nano/microcellulose, chitosan, alginate, gelatin, polyethylene glycol (PEG), poloxamer (a triblock amphiphilic PEG-PPG-PEG copolymer), carbomer (polyacrylic acid), poly-NIPAM (poly-N-isopropylacrylamide), polyacrylamide, gellan gum, and polylactide, each of which has its particular physical-chemical properties [18,19,20,21,22,23,24,25].

Bacterial nanocellulose (BNC) is a biopolymer with extremely appealing properties and applications, especially in its native, never-dried state, such as biocompatibility, high hydrophilicity, flexibility, transparency, high mechanical strength and chemical stability, and high surface area [19,20,26,27,28]. Never-dried BNC (NDBNC) has a net-like structure formed by nanofibrils of 40–60 nm in diameter and its purity is higher than that of plant cellulose. The BNC chains have a high polymerization degree with up to 8000 cellobiose units arranged predominantly in an Iα one-chain triclinic allomorph that is more flexible than Iβ, with its two-chain monoclinic structure, a high crystallinity up to 90–95%, and a structural bound water up to 99% [29]. Moreover, NDBNC can entrap different bioactive compounds and enable their delivery to the target site [30,31,32,33].

Poloxamer 407 (PX) is a FDA-approved nontoxic medical-pharmaceutical ingredient, chemically described as an amphiphilic triblock copolymer of poly(ethylene oxide)/poly(propylene oxide)/poly(ethylene oxide) or poly(ethylene glycol)/poly(propylene glycol)/poly(ethylene glycol), with the general formula PEG_101_/PPG_56_/PEG_101_ and the average molecular weight of 12,600 Da [34]. It has 70% hydrophilic character, which is represented by the marginal PEG chains, while the PPG core is hydrophobic. PX is very soluble in water up to high concentrations of 15–30%, at which its thermo-sensitive gelation behavior can be exploited. Depending on concentrations, poloxamers have a lower critical solution temperature (LCST) below the body temperature, meaning that they are liquid below LCST and contract as gels above LCST, being suitable for injectable targeted delivery of bioactive compounds [35].

Besides poloxamer and cellulose derivates, chitosan (CS), which is a linear polysaccharide, is also frequently used in various formulations for temperature- and pH-sensitive hydrogels with biomedical applications [17,35,36,37,38,39,40]. Several studies showed the potential of chitosan as a protective agent against pathogenic bacteria, but it also has anti-inflammatory properties, stimulating the wound-healing process [41,42,43,44,45].

Inulin is a polysaccharide composed mainly of fructose and is among the compounds accepted as a prebiotic by the International Scientific Association for Probiotics and Prebiotics (ISAP). Prebiotics are components of food that act as a growth substrate for the health-promoting living microorganisms (probiotics) [46,47,48].

The antimicrobial activity of thyme essential oil (TEO) against several pathogenic species is well known, and several formulations that include TEO for the treatment of vaginal infections have been developed [49,50,51,52,53]. Several other studies have shown the antifungal effect of *Vitis vinifera* seed extract [54,55]. The lactic acid produced by the *Lactobacillus* genus from vaginal microbiota maintains a slightly acidic pH in the cytosol, at which pathogenic species cannot proliferate. Additionally, an increased level of lactic acid leads to the death of pathogenic species by activating a cascade of biochemical and immunological processes for maintaining a balance between symbiotic commensal species [56]. Citric acid has gained attention in the pharmaceutical and cosmetical field as a remarkable excipient due to its useful properties, such as biocompatibility and biodegradability, being a pH adjuster, crosslinker, and chelating and stabilizing agent, and also being a natural compound [57,58]. Regarding the biological effects possessed by the extracts from different parts of *Opuntia ficus-indica*, several studies showed its antioxidant, antibacterial, and anti-inflammatory activities, as well as its tissue regeneration capacity [59,60,61,62].

Mucoadhesion is an important property for hydrogels applied on healthy, sensitive, or wounded skin, or on soft tissues such as the oral and nasal cavities, vulva, and vagina [63,64,65,66,67,68]. The vagina is an extremely sensitive area since there are many factors that can destabilize the homeostasis of the vaginal microbiota, such as poor hygiene, puberty- or pregnancy-induced hormonal changes, sexual activity, and various drugs prescribed for certain medical conditions [69]. The topic of this study relates to new alternative hydrogels as promising medical devices for treating and/or preventing common infections affecting more than 70% of adult women, such as bacterial vaginosis (BV), aerobic vaginitis (AV), or vulvovaginal candidiasis (VVC) [13,70,71], while considering also the protection or re-balance of vaginal microbiota.

Binary PX-CS, BNC-CS, or BNC-PX formulations have been reported previously with promising properties in the biomedical field [72,73,74,75]. There are some previous reports on hydrogel combinations of poloxamer and chitosan for vaginal applications [76,77,78,79]. Formulations of poloxamer-(nano)cellulose/cellulose derivatives or CS-(nano)cellulose/cellulose derivatives have been less studied, although nanocellulose formulated with gellan and tetraethyl orthosilicate (TEOS)-modified nanocellulose scaffolds have been proposed as vaginal fluconazole delivery systems [32,80]. The advantage of ternary hydrogels over binary hydrogels might reside in increased stability, mucoadhesion, swelling, or drug delivery properties by accumulation [81,82,83,84,85] or even synergism [86,87,88] of the particular features of individual polymers. To the best of our knowledge, no poloxamer-bacterial nanocellulose or poloxamer-bacterial nanocellulose-chitosan hydrogel formulations have been reported for vaginal applications. Moreover, most of the studies involving hydrogels investigated loading and delivery of synthetic known drugs, usually with hormone balancing or antimicrobial properties. Hydrogels incorporating natural compounds/extracts for the vagina have received little to no attention.

The aim of the study was to investigate the structural and functional properties of two Kombucha BNC-PX formulations and a triple-polymeric hydrogel matrix based on Kombucha BNC, CS, and PX loaded with bioactive extracts, in order to support the vaginal microbiota homeostasis. To the best of our knowledge, these types of formulations are reported for the first time in the literature.

## 2. Results

### 2.1. Structure and Physical-Chemical Properties of the Hydrogels

The formulation of the hydrogels was centered around flexible fibrillar bacterial nanocellulose (BNC) and Poloxamer 407 (PX) as an amphiphilic binder between hydrophilic and hydrophobic compounds, and as a thermo-responsive polymer. Two hydrogels were obtained with 0.4% BNC, 15% PX, and different biocompounds following the recipes presented in Section 4.2, together with a third ternary hydrogel based on 0.4% BNC, 5% PX, and 3% chitosan (CS).

Three types of hydrogels were obtained with the following pH: H1 hydrogel, pH = 4.31; H2 hydrogel, pH = 4.35; H3 hydrogel, pH = 4.80. The description of the hydrogels is provided in Materials and Methods. The appearance of the two bases and three hydrogels is presented in Figure 1.

The base B1,2 had a homogeneous, milky-white aspect, with low to medium viscosity, similar to a body lotion, while the base B3 was beige-translucent, with an aerated aspect, and significantly more viscous than B1,2; its morphological aspect was attributed to chitosan. H1 appeared as a homogeneous, milky-white formulation with medium viscosity like that of a soft body cream. H2 had a yellow-brown appearance, with visible particles of *Opuntia ficus-indica* that give a particular morphological aspect to H2, and a medium viscosity similar to that of H1. H3 appeared as a homogeneous, milky-white formulation with high viscosity and the consistency of a body butter. All the hydrogels showed a good homogeneity and pleasant odor mainly due to the thyme essential oil.

#### 2.1.1. Microscopic Structure of Hydrogels via TEM

Transmission electron microscopy results are presented in Figure 2 for hydrogel H1 and for hydrogel H3, and evidenced PX micelles of around 25–50 nm in diameter either free or arranged as nano-pearls on the BNC nanofibrillar chains of 10–50 nm in diameter. The chitosan from H3 appears in Figure 2b,d as a shadow surrounding the bacterial nanofibrils, in contrast with the absence of this shadow in Figure 2a,c for the hydrogel H1 based on BNC and PX.

#### 2.1.2. Molecular Interactions in Hydrogels by FTIR Spectroscopy

The FTIR spectra of individual polymers BNC, CS, and PX are presented in Appendix A and the assignment of bond vibrations is presented in Appendix A. In Figure 3a, the spectra of the binary and ternary hydrogel bases are overlaid, i.e., the B12 base for H1 and H2 consisting of 0.4% BNC and 15% PX, and the B3 base containing 0.4% BNC, 5%PX, and 3% CS as the base for the hydrogel H3. In Figure 3a, the FTIR spectra of the individual polymers and that of 3% chitosan solution in 1% acetic acid (CSAcAc) are also included. The spectrum of CSAcAc differs from the one of the unprotonated CS mainly in the amide II and III bands around 1543 cm^−1^ [89,90] and 1400–1375 cm^−1^, respectively, assigned to the ionic ammonium-carboxyl N^+^COO^−^ system. The spectrum of the B1,2 base mainly resembles the PX spectrum due to the high PX concentration, with small shifts of the C-H band [91] from 2880 cm^−1^ in PX to 2882 cm^−1^ in B1,2, and of the C-O band from 1096 cm^−1^ to 1097 cm^−1^ [91,92]. The IR shifts towards higher wavenumbers are generally assigned to stronger interactions and higher stability. Another small difference in B1,2 compared to PX is the small peak at 669 cm^−1^ assigned in Appendix A to in-plane rocking vibrations of heterocyclic C-H in gluco-pyranosic rings of BNC. In Figure 3a it can also be observed that the B3 spectrum resembles the CSAcAc spectrum, evidencing that the 3% CS influence is dominant over that of the 5% PX and 0.4% BNC. There are small shifts towards higher wavenumbers, from 1543 cm^−1^ and 1024 cm^−1^ in CSAcAc to 1547 cm^−1^ and 1030 cm^−1^ in B3.

The FTIR spectra of the final hydrogels H1, H2, and H3 are presented in Figure 3b in comparison with the corresponding bases B1,2 and B3 and with the components citric acid, lactic acid, thyme essential oil, *Vitis vinifera* seeds extract, *Opuntia ficus indica* powder, and inulin. Citric acid is present in all hydrogels at 3% concentration and induces the characteristic citrate band around 1732 cm^−1^ for H1 and H2, compared to around 1721 cm^−1^ for H3, together with a small peak around 1207 cm^−1^. The band around 1715 cm^−1^ is characteristic of -COOH in lactic acid, which might contribute as lactate to the hydrogel band around 1732 cm^−1^ for H1 and H2, compared to 1721 cm^−1^ for H3. Moreover, the methyl band at 1454 cm^−1^ in lactic acid gives a particularly different vibration than the methyl groups in PX at 1466 cm^−1^; these two signals are best visible in hydrogel H2, as evidenced by the encircled area in Figure 3b. Thyme essential oil presents C-H absorption bands at 2959, 2924, and 2868 cm^−1^ specific for the aliphatic chains of fatty acids [93], while in the hydrogels these signals are significantly decreased due to the reduced concentration used, namely, 0.5%. The C-H band is dominated in hydrogels by the PX chains with the main peak at 2880 cm^−1^ in PX alone and H2, shifted to 2876 cm^−1^ in H1, and to 2878 cm^−1^ in H3. The C-C-O band within 1460–1250 cm^−1^ and the C-O-C bands around 1100, 962, and 841 cm^−1^ from PX [91,92] are better visible in H1 and H2 compared with H3. The hydro-glycero-alcoholic extract of *Vitis vinifera* seed is characterized mainly by the -OH absorption bands around 3300 ± 200 cm^−1^, 1645 ± 90 cm^−1^, and 1040 ± 100 cm^−1^, while the reduced amount of 0.5% does not seem to influence the hydrogels’ spectra. The amount of 3% inulin mainly influences the hydrogen bond area in the region 3600–3000 cm^−1^, together with the other compounds, but it also contributes in the region 1000–900 cm^−1^. *Opuntia ficus-indica* is added at 0.1% concentration only in H2 and this small amount does not influence the FTIR spectrum of H2. Chitosan was used only in H3 and a few characteristic bands appear around 1572 and 1030 cm^−1^, specific to C-N-H vibrations in amide II and the C-O-C glycosidic bond, respectively [89,90].

#### 2.1.3. Rheological Properties of the Hydrogels

The rheology experiments of hydrogels were performed in four different modes with hysteresis (up and reverse variations mean increases and decreases, respectively, in shear or temperature) and results are presented in Figure 4; each was relevant for particular aspects of hydrogel behavior under stress, temperature, and time. “Oscilo sweep”, or oscillation frequency sweep, is a mode used to test the frequency and time dependence of the viscoelastic properties, where the high frequencies correspond to short time scales and the low frequencies to long time scales.

In the first mode, i.e., oscillation/frequency sweep, the frequency is varied linearly while the strain or stress amplitude of the sinusoidal deformation remains constant. The frequency sweep presented in Figure 4a–c allows the evaluation of storage modulus G′, loss modulus G″, and complex viscosity η* in the angular frequencies ω range of 0.1–100 rad/s. For H1 and H2, the storage modulus G′ is larger than the loss modulus G″, which means that they are predominantly elastic (solid-like). H3 behaves like a viscoelastic fluid, with the G′ curve crossing the G″ curve around an angular frequency ω of 1 rad/s and G′ = G″ = 65.5 Pa.

The phase angle variation δ (°) at 25 ± 1 °C is presented in Figure 4a–c, being a relevant parameter for the viscoelastic fluids and their response during application. Phase angle δ represents the shifting angle between the input strain and the output stress, taking values between 0° and 90°, where δ = 0° corresponds to a purely elastic response (Hookean solid) and δ = 90° corresponds to a purely viscous response (Newtonian liquid). For H1 in UP mode, the phase angle δ increases from 22.8° at 0.1 rad/s to 27.5° at 0.16 rad/s, and further decreases to 20.1° at 4 rad/s and stabilizes around 21.1° at 100 rad/s, while in the reverse (R) mode, δ increases from 18.9° at 100 rad/s to 35.8° at 0.1 rad/s. The increase in phase angle suggests the fluidization of the hydrogel during shearing, while the decrease suggests the stabilization as a viscoelastic gel. For H2 in the UP mode, the phase angle δ decreases from 17.2° at 0.1 rad/s to a minimum of 13.2° at ω = 0.6 rad/s and increases to 30.3° at 100 rad/s, while in the R mode, δ decreases from 27.9° at 100 rad/s to a minimum of 14.4° at ω = 2.5 rad/s and further increases up to 19.6° at 0.1 rad/s. For H3 in the UP mode, the phase angle δ decreases from 55.9° at 0.1 rad/s to 23.1° at 100 rad/s, while in the R mode, δ increases from 23.1° at 100 rad/s to 54.4° at 0.1 rad/s. An additional parameter presented in Figure 4a–c and related to the possible perception of hydrogels is the stiffness k. The stiffness is relatively constant for all three hydrogels up to 25–30 rad/s, while after 30 rad/s it increases rapidly, suggesting the stability of the formulations at high shear rates, while the similarity of the reverse curves confirms the stability at the application spot and the gels’ relaxation towards the initial low stiffness.

In Figure 4d–f, the rheological curves were fitted with the main rheological models and compared by the best fitting method for viscosity and stress. The Carreau model [94,95] best describes the viscosity of all hydrogels, while the zero-rate viscosity parameter indicates a higher viscosity for H2, followed by H1 and H3. Additionally, the hydrogels H1 and H2 behave like shear thinning Hershel–Bulkley structural fluids [96,97] with a yield stress σ_y_; this is the stress value from which the hydrogel starts to flow, and is higher for H2 (4.6 Pa) than H1 (3.8 Pa). The yield stress is a time-dependent property and correlates with the storage stability by preventing sedimentation; it also correlates with an increased shelf life and with good stability at vibrations during transportation [98].

H1 has an initial (UP) yield stress of 3.8 Pa and a reverse (R) yield stress of 1.1 Pa according to the results from the Hershel–Bulkley model fitting, while H2 has an initial yield stress of 4.6 Pa and a reverse yield stress of 0.9 Pa. These values suggest that H1 and H2 hydrogels behave like a diluted body lotion; a commercial example with a pump application has a value of around 27 Pa for the yield stress [98]. Hydrogel H3, although it does not have a yield stress according to the results from the best fit correlation, behaves like a pseudoplastic (shear thinning) structured fluid with good stability induced by chitosan, as can be seen from the overlaying of the UP and R curves for storage and loss modulus. Moreover, the thixotropy index shows that H1 and H2 are thixotropic fluids, with H1 being more thixotropic than H2. H3 behaves differently, showing slightly rheopectic (anti-thixotropic) behavior, the reverse stress being slightly higher than the initial, UP, stress, and similarly for viscosity. The evaluation of all the fitting models is presented in Appendix A.

The axial mode (Figure 4g–i) can be seen as an adhesion test or tack test since it deforms the sample in a vertical motion of the geometry plate with a constant speed of 20 µm/s in our case, while recording the axial force F(N) that opposes the movement. The maximum axial force and the energy of adhesion can be determined with this test. The maximum axial forces for H1, H2, and H3 were 0.338 N, 0.377 N, and 1.695 N, and the maximum adhesion times were 1.06 s, 1.05 s, and 3.06 s, respectively. The adhesion energy can be seen as the work of adhesion or the energy to separate 1 m^2^ of joined materials for the distance of separation, and it is expressed in Nm/m^2^ or J/m^2^. The work distance can be interpolated in the axial force graphic at the value of null axial force (or last negative value of F), and the corresponding determined distances (d) for H1, H2, and H3, minus the 1 mm gap, were 3.643, 4.244, and 6.781 mm. The contact surface is the 40 mm geometry (cylinder) area, meaning S = 1.257 × 10^−3^ m^2^. The resulting adhesion energies (A_E_) were 1.124, 1.273, and 9.144 J/m^2^ for H1, H2, and H3, respectively. Moreover, the axial force curve was fitted with an exponential function, for which the most relevant parameter is the power coefficient c, further interpreted as the speed of detachment. In this view, the highest adhesion character is correlated with the lowest speed of detachment, namely, c = 13.28 for H3, compared with c = 20.46 for H1 and c = 17.55 for H2, confirming the previously estimated adhesive energy order.

The temperature influence is presented in Figure 5a–c in an oscillatory experiment at constant angular frequency of 10 rad/s with the temperature increasing and decreasing in a ramp from 10 °C to 40 °C and back, with the aim to determine the gel transition temperature for the hydrogels. From Figure 5a (H1) and Figure 5b (H2), an increase in G′, G″ and tan(δ) of around 18 °C for H1, compared to 22 °C for H2, can be observed, suggesting their gel transition temperature. The hydrogel H3 apparently did not have a gel transition temperature, but it had very good stability with temperature, as can be seen from the similar UP and R curves in Figure 5c. The stiffness parameter started to linearly increase after 18 °C for H1, and after 22 °C for H2, confirming therefore their sol–gel transition temperatures. On the contrary, the stiffness of H3 linearly decreased with increase in temperature, but the k values are comparable with those of the other two hydrogels, i.e., within 0–0.2 N.m/rad, in the decreasing order k_H1_ > k_H3_ > k_H2_. The temperature influence at constant flow rate presented in Figure 5d–f confirms the sol–gel transition temperatures for H1 and H2 at the inflection point of the S-shaped curves of both viscosity and stress, and additionally presents the maximum values and the plateau between 25 and 40 °C. The viscosity of the hydrogels between room and body temperature does not change significantly for all hydrogels.

#### 2.1.4. X-ray Diffraction of Bases and Loaded Hydrogels

X-ray diffraction (XRD) analyses were performed on solid individual components of hydrogel bases B1,2 and B3, i.e., BNC, PX, and CS, and also on final hydrogels and their other components, i.e., inulin, citric acid, and *Opuntia*; results are presented in Figure 6. PX has specific diffraction peaks at 2θ = 19.12° and 23.2°, as can be seen in Figure 6a; similar peaks were previously reported at 2θ = 19.2° and 24.0° [99]. Chitosan is a semi-crystalline biopolymer with a diffraction pattern consisting of two main peaks around 2θ 9.5° and 19.94°, as can be seen in Figure 6a; similar values of 10° and 20° for chitosan were previously reported [100]. BNC obtained from Kombucha membrane was obtained and characterized as in our previous work [26], and here we only briefly mention that it contains cellulose Iα, Iβ, and amorphous cellulose convoluted in the three main peaks around 14.42°, 16.72°, and 22.7°, visible in Figure 6a. The base B1,2 containing BNC 0.4% and PX 15% is dominated by the PX diffraction peaks, which are slightly shifted towards BNC peaks, suggesting interactions between the BNC and PX chains. The base B3 containing BNC 0.4%, PX 5%, and CS 3% showed a decreased intensity of all peaks, an apparent absence of initial chitosan peaks, and a decreased crystallinity of only 42% compared to the individual polymers that have much higher crystallinity (PX—63%, CS—83%, BNC—57%). The absence of chitosan bands suggests a high degree of chitosan protonation due to the acetic acid present, structural rearrangements, and strong interactions with the other B3 polymers.

The final hydrogels H1, H2, and H3 presented in Figure 6b showed new peaks specific to the added components and to particular interactions. The first observation is the almost complete absence of PX peaks in H3; only a small specific peak around 19° is visible, which suggests a strong interaction of PX with the new biocompounds inside the hydrogel H3. The BNC presence is suggested by the small diffraction peaks around 16.6° and 21.8° for cellulose Iα, as can be seen in Figure 6a. Chitosan still does not show any of its distinctive crystalline peaks. Inulin is present in the same amount in all three hydrogels and it increases the amorphous region around 18° in all hydrogels. *Opuntia ficus-indica* is present only in H2 in a small amount of 0.1%, inducing the small peak at 27.92° in H2. A new and unchanged peak appears at 12.12° in all three hydrogels, H1–H3, compared to the bases B1,2 and B3, which, together with the small peak around 8°, might be assigned to a citrate compound or citric acid interaction within the hydrogels, as we will further discuss.

### 2.2. Hydrogel Interaction with Mucin

#### 2.2.1. Quantitative Evaluation of Hydrogel–Mucin Interaction

Due to the interference of certain compounds of the hydrogel composition with the PAS reagent, and thus the inability to obtain reliable results by subtracting the concentration of free mucin from the initial mucin concentration, the mucin binding efficiency of the hydrogel bases was determined. At a base/mucin ratio of 15 (*w*/*w*), the mucin binding efficiency was about 50%, with no significant differences between B1,2 and B3. By increasing the base/mucin ratio to 44.8 (*w*/*w*), the mucin binding efficiency increased to 59.5 ± 3.0% and 61.4 ± 1.5% in the cases of B1,2 and B3, respectively, with a marginally statistically significant difference between the two bases (Figure 7).

#### 2.2.2. SEM Microscopic Structure of Hydrogel Interaction with Mucin

The SEM micrographs of the hydrogels and hydrogel–mucin are shown in Figure 8. H1 has a compact appearance, with abundant micellar structures (Figure 8a), and after the contact with mucin it becomes a porous, fibrillar mesh structure (Figure 8b). Rearrangements occur in the structure of the H2 hydrogel compared with H1, which is a rather fibrillar but also a porous structure (Figure 8c). After the contact with mucin, the formation of a network-like structure can be observed, with plenty of thin fibers and small pores (Figure 8d). H3 has a different structure and behavior compared to H1 and H2. Figure 8e shows that H3 presents an ordered lace-like structure, which after contact with mucin becomes more compact and the lace-like formations shrink (Figure 8f). The H3–mucin structure partially resembles the H2–mucin structure.

#### 2.2.3. Molecular Interactions of Hydrogels with Mucin via FTIR Spectroscopy and XRD

The molecular interactions between H1, H2, and H3 hydrogels and 3.5% mucin aqueous solution were investigated via FTIR spectroscopy and are shown in Figure 9a. The main spectral changes appear in the amide bands I and II, as can be seen by the shape variation of the specific bands highlighted with a pink shadow in Figure 9a. The mucin amide I band around 1630 cm^−1^ specific for amino acids and proteins convolutes with the corresponding hydrogel bands, and the amide II band around 1520 cm^−1^ in mucin appears distinctively in the hydrogel–mucin systems. The band at 1227 cm^−1^, which could have a contribution from amide III and/or a C-O vibration, is shifted and reduced by convolution with the bands from the hydrogels. The glycosidic band at 1034 cm^−1^ in mucin is reduced and melts within the bands of hydrogels. Changes also occur in the spectra of hydrogels. The band at 1726 cm^−1^, characteristic to the COOH group from lactic and/or citric acid, seems to be slightly reduced in the presence of mucin. The main band at 1103 cm^−1^ in the saccharides region became sharper upon hydrogel mixing with mucin. Some other changes occur in the region 1350–1250 cm^−1^, the sharp peaks from PX and/or citric acid becoming more intense in the presence of mucin (red circles). The bands at 1030 and 1060 cm^−1^, from BNC/CS and possibly *Vitis vinifera*, were reduced and intensified, respectively (see the black arrows in this region). In the case of H3, the changes in hydrogel bands are more visible than those in H1 and H2. If H3 had the maximum peak in the glycosidic region at 1030 cm^−1^ (BNC/CS), upon mixing with mucin, the peak from PX at approx. 1103 cm^−1^ becomes the most intense and the spectrum is changed significantly. Some changes also occur in the region 1000–900 cm^−1^, i.e., changes in some band intensities and shift of the band 937 cm^−1^ to 944 cm^−1^; this band probably comes from inulin and maybe thyme essential oil (red circle and black arrow). These changes indicate interactions between the hydrogels and mucin that involve both the glycosidic and peptide bonds of mucin and the biopolymers and other biomolecules in hydrogels.

The interaction between hydrogels and mucin was also investigated using XRD, and results are presented in Figure 9b. The diffractograms of hydrogels with and without mucin are partially overlaid in order to evidence the main effect, i.e., the decrease in the amorphous band together with the disappearance of many small peaks previously assigned to various bioactive components. Crystallinity increases significantly and the crystalline bands characteristic to poloxamer are more visible than in the hydrogels in the absence of mucin.

#### 2.2.4. Rheological Studies of Hydrogel–Mucin Complex

The rheological experiments with 50% dilution of the H1, H2, and H3 hydrogels with 3.5% mucin aqueous suspension presented in Figure 10 evidenced a general decreasing viscosity of the mixed systems compared with the initial hydrogels. The dilution of topical hydrogels in vivo is estimated to be up to 67–80% remaining hydrogel [101], so the 50% dilution considered in our tests is an extreme situation.

Both the oscillatory and flow experiments presented in Figure 10a–c and Figure 10d–f, respectively, evidence a stabilization effect of the mucin on the hydrogel flow at medium shear rates; the UP and R curves are closer to each other and, therefore, the thixotropy index is smaller than for the unbound hydrogels. At angular rates higher than 25–40 rad/s, comparable to walking activities, the storage modulus starts to decrease for all hydrogels, suggesting the loss of the elastic character and, therefore, the disruption of hydrogels by the mucin interaction and shear. In Figure 10d–f, the decrease in the thixotropy index, the decrease in the yield stress values, and the change to Casson as the best fitting model [102], specifically for lower yield stress fluids, can be observed.

The axial test also evidenced a significant decrease in the maximum axial forces, which for H1 + Mu, H2 + Mu, and H3 + Mu, minus the 1 mm gap, are 3.445, 2.965, and 3.886 mm, respectively. The resulting adhesion energies of the hydrogels H1, H2, and H3 in the presence of mucin are 0.249, 0.167, and 0.343 J/m^2^, respectively. If we recall the adhesion energies for the hydrogels without mucin, of 1.124, 1.273, and 9.144 J/m^2^, respectively, we can determine a so-called binding energy or cohesion energy [103,104,105] between hydrogels and mucin as a difference, i.e., 0.875, 1.106, and 8.801 J/m^2^ for the three hydrogels, H1, H2, and H3, respectively. This indicates a very high cohesion energy in the ternary hydrogel. The adhesion times remain relatively similar, at 1 s, for H1 + Mu and H2 + Mu compared with those of H1 and H2, and for H3 + Mu the adhesion time decreases from 3 s to 2 s. The exponential functions showed an increase in the detachment speed c of the hydrogel–mucin systems compared with the hydrogels alone, which showed a decrease in the adhesion, while H3 remained the most adhesive even after the contact with mucin, with the lowest detachment speed of c = 44.53.

The influence of temperature on the hydrogel–mucin systems is evidenced in Figure 11a–c at a constant oscillatory rate, and in Figure 11d–f at a constant flow rate. The hydrogel–mucin systems do not show a sol–gel transition in the UP temperature ramp, but H1 + Mu shows a gel–sol transition of around 35 °C on the R (decreasing) temperature curve. The viscosity and stress variations of the hydrogels after the contact with mucin at body temperature show only small variations, of a maximum of 0.1 Pa·s, compared to 0.6 Pa at room temperature.

### 2.3. Biocompatibility and Bioactivity of Hydrogels

#### 2.3.1. Biocompatibility Assay

The MTT assay showed that the viability of the NCTC cell line following exposure to serial hydrogel concentrations ranged from 72.1 ± 1.3% to 119.62 ± 2.7% when compared to untreated cells (Figure 12). With the exception of the 100 µg/mL hydrogel dose, which was slightly cytotoxic, the other tested concentrations were biocompatible and most of them had the potential to stimulate cell proliferation. The lowest concentration of hydrogel, i.e., 12.5 µg/mL, led to the highest increase in the number of metabolically active cells (119.1 ± 2.8% after H1 treatment, 118 ± 3.2% after H2 treatment, 119.6 ± 2.7% after H3 treatment, with respect to the control C), and there were no significant differences between the three hydrogel systems. The 25 µg/mL hydrogel concentration induced a slight decrease in cell viability for all the tested hydrogels compared to 12.5 µg/mL hydrogel, the decrease being the smallest in the case of H2 (110.6 ± 1.9% for H1, 114.9 ± 2.0% for H2, and 111 ± 1.44% for H3). At 50 µg/mL hydrogel, the number of metabolically active cells further decreased compared with 25 µg/mL hydrogel, but only the H1 treatment induced cell viability below the control of untreated cells C (95.5 ± 3% upon the H1 treatment, 105 ± 2.7% upon the H2 treatment, and 110.5 ± 1.2% upon the H3 treatment).

#### 2.3.2. Prebiotic Activity of the Hydrogels

At 24 h after treatment with various concentrations of hydrogel, there were no significant differences between the treatment with hydrogel and the positive control (C+) on the growth of *L. reuteri* (Figure 13a). At 48 h after treatment, at the lowest H1 concentrations tested, i.e., 12.5 and 25 µg/mL, there was a slight inhibition of *L. reuteri* growth, which was marginally statistically significant (97.4 ± 0.5% for 12.5 µg/mL H1, and 98.3 ± 0.7% for 25 µg/mL H1). By increasing the H1 concentration to 50 and 100 µg/mL, the percentage of *L. reuteri* growth reached the C+ level after 48 h. Some concentrations of hydrogels H2 and H3 induced a significant increase in the *L. reuteri* growth after 48 h incubation. The treatment with 50 µg/mL H2 induced a marginally significant increase in *L. reuteri* growth (105.9 ± 1%), and 100 µg/mL induced a significant increase (114 ± 0.9%) compared to C+. The lowest tested concentrations of H2 did not induce changes in *L. reuteri* growth compared to C+. In the case of H3 there was a marginally significant difference at the highest concentration tested (103.6 ± 0.8%) compared to C+, the other concentrations having no effect on *L. reuteri* growth (Figure 13b). After 72 h of treatment, a similar trend as that after 48 h of treatment was observed, but there was a slight decrease in *L. reuteri* growth for all the tested concentrations (Figure 13c). The only significant prebiotic effect after 72 h was obtained at 50 µg/mL H1 (105 ± 1.5%) and at 100 µg/mL H2 (111.3 ± 3.4%).

Regarding the influence of different concentrations of hydrogel on the growth of *L. plantarum* at 24 h after the treatment, a marginally significant growth inhibition (99.6 ± 0.3%) was noted at the lowest concentration of H1. A marginally significant growth promotion (101.5 ± 0.2% and 101.7 ± 0.4%) was observed at the highest concentration (100 µg/mL) of H1 and H2, respectively (Figure 14a). At 48 h after the treatment (Figure 14b) and 72 h after the treatment (Figure 14c), there were no significant differences between the hydrogels and C+.

#### 2.3.3. Antimicrobial Activity of the Hydrogels

The results presented in Table 1 and Appendix A highlight the significant antimicrobial activity of the tested hydrogels. At the low dose (25 µL), the most effective hydrogels against *E. coli* proved to be H1 (1.130 ± 0.080 cm) and H2 (1.010 ± 0.004 cm), with the antibacterial activity of the hydrogel H3 being slightly lower (0.740 ± 0.008 cm) compared to the other two hydrogels. In contrast, the antifungal activity of H3 (1.010 ± 0.010 cm) was significantly higher compared to that of H1 (0.280 ± 0.010 cm) and H2 (0.310 ± 0.010 cm) when tested on *C. albicans*. At the high dose of hydrogel (100 µL), all formulations were effective against the selected microbial strains. In the case of H3, the antibacterial activity had a partial contribution from the solvent (control C3), probably induced by the acetic acid used for chitosan solubilization. By diluting the hydrogels with double-distilled water in a ratio of 2:1 (hydrogel: water–*v*/*v*), the *E. coli* inhibition increased for all the tested hydrogels. However, in the case of *C. albicans*, H1 and H2 lost their ability to produce strain inhibition, and the diameter of the inhibition zone in the case of H3 was slightly reduced upon dilution compared with undiluted hydrogels.

We next used 50 µg/mL hydrogel suspensions for further tests of antimicrobial activity, as this concentration had both the potential to stimulate cell proliferation and to promote lactobacilli growth. Using different inoculations of bacterial suspensions (between 1.5 × 10^7^ and 1.5 × 10^1^/well), inhibition of *E. coli* growth was observed at 12 h and 24 h after the treatment, the inhibition increasing as the concentration of bacterial cells decreased. The samples belonging to the same bacterial density were analyzed separately from the statistical significance point of view. In the case of H1, the inhibition started from 4.3 ± 0.4% at 1.5 × 10^7^ bacterial cells and reached 37.2 ± 3.6% at 1.5 × 10^1^ bacterial cells after 12 h of incubation. In the case of H2, the growth inhibition was 9.3 ± 1.1% at the highest bacterial cell density, and it reached 99.0 ± 0.1% at the lowest bacterial density. Although H3 was not effective at a higher cell density, at 1.5 × 10^1^ bacterial cells the inhibition was similar to that of H2, inducing 96.5 ± 0.2% cell growth inhibition (Figure 15a).

After 24 h of incubation, the inhibitory capacity of H2 decreased significantly and became lower than that of H1. The dependence of the inhibitory percent on cell density was similar to that after 12 h of incubation. In the case of H1, the inhibition started from 7.6 ± 0.3% at 1.5 × 10^7^ bacterial cells and reached 23.9 ± 5.8% at the lowest bacterial density. H2 inhibited the growth of *E. coli* by 1.01 ± 0.06% at 1.5 × 10^7^ bacterial cells, and by 14.8 ± 2.2% at the lowest bacterial density. In the case of H3, the level of inhibition at 24 h remained approximately the same as that observed at 12 h, reaching 97.6 ± 0.09% at 1.5 × 10^1^ bacterial cells (Figure 15b). The bacterial cell density of 1.5 × 10^0^ was used as a negative control, as there was no bacterial growth even in the absence of hydrogel.

After determining the growth inhibition, the same samples were investigated for biofilm formation (Table 2). Inhibition of bacterial biofilm was observed after treatment with H1 at 1.5 × 10^7^ and 1.5 × 10^6^ bacterial cells, reaching approximately 60% inhibition compared to the positive control. At 1.5 × 10^5^, 1.5 × 10^4^, and 1.5 × 10^3^, the formation of biofilm was not observed in the presence of H1, with the antibiofilm activity being 100%. The treatment with H2 led to complete inhibition of bacterial biofilm formation in the case of all bacterial cell densities. In the case of H3, a reduction of approximately 24% was observed at the highest bacterial cell density. The maximum biofilm inhibition, 38%, was induced by H3 at 1.5 × 10^3^bacterial cell density. At 1.5 × 10^2^ and 1.5 × 10^1^, no biofilm was formed even at the level of the positive control.

In the case of *C. albicans*, the inhibitory effect of the 50 µg/mL hydrogel suspensions increased as the density of microbial cells decreased, similar to the case of *E. coli* (Figure 16).

From 1.5 × 10^2^ to 1.5 × 10^0^ cell density, there was no growth of *C. albicans* even in the positive control. After 12 h, following the treatment with H1, the inhibition of *C. albicans* growth was 2.8 ± 0.2% at the highest microbial density (1.5 × 10^7^), reaching 100% inhibition at 1.5 × 10^3^ microbial cells. In the case of H2, the inhibition started from 3.5 ± 1.1% at 1.5 × 10^7^ microbial density and reached 99.06 ± 2.09% at 1.5 × 10^3^. Following the treatment with H3, the inhibition at the highest microbial density was 4.9 ± 2.6%, reaching 94.7 ± 2.8% at 1.5 × 10^3^ (Figure 16a). After 24 h, at microbial density between 1.5 × 10^7^ and 1.5 × 10^4^, no significant inhibition of the growth of *C. albicans* was observed. At 1.5 × 10^3^, an inhibition of approximately 98.8 ± 0.2%, 100 ± 0.4%, and 98.9 ± 0.3% in the case of H1, H2, and H3, respectively, was obtained (Figure 16b).

## 3. Discussion

As mentioned, we report for the first time, to the best of our knowledge, two binary hydrogels (H1, H2) based on never-dried bacterial nanocellulose (NDBNC) and poloxamer (PX) and one ternary hydrogel containing NDBNC, PX, and chitosan (CS) for vaginal applications. The hydrogels were loaded with bioactive compounds such as inulin, thyme essential oil, *Vitis vinifera seed* extract, lactic acid, citric acid (H1, H2, H3), and *Opuntia ficus-indica* powder (H2). Addition of compounds to B3 induced a visible change in the morphological aspect of H3 compared to that of B3 (Figure 1).

Several studies demonstrated the potential of inulin for vaginal health by enhancing the growth of several lactic acid bacteria (LAB) and inhibiting the growth of *C. albicans* [106,107]. By investigating the antimicrobial effect of thyme essential oil against 14 strains of *Candida*, the MIC value ranged from 0.03 to 8% (*v*/*v*) [108]. *Vitis vinifera* seed hydroalcoholic extract has been shown to inhibit *C. albicans* at concentrations between 5.7 and 20.2 mg/L minimal inhibitory concentration (MIC), depending on the cultivars, the effect being due to the its high content of polymeric flavan-3-ols [55]. In another study, inhibition of *C. albicans* was observed at a concentration of 0.3 g/mL MIC of *Vitis vinifera* seed aqueous extract [54]. The significant difference in MIC between the two studies could be determined by the type of cultivar, agronomic conditions, and/or the method of extraction. Lactic acid can lead to the production of neutrophil gelatinase-associated lipocalin (NGAL) by the vaginal epithelial cells [109,110], production of H_2_O_2_, which inhibits the catalase-negative anaerobic species [111], and production of bacteriocins [112]. However, it can also stimulate the anti-inflammatory cytokines and DNA repair by inhibition of histone deacetylase activity [113] in order to reestablish the vaginal microbiota homeostasis. Therefore, the addition of lactic acid in our formulations had the aims of both lowering the pH and providing its other beneficial properties until reestablishing the vaginal microbiota homeostasis. Considering the above observations, we added different concentrations of bioactive compounds in our binary and ternary formulations in order to complement each other and improve the biological efficacy of the hydrogels for the treatment of vaginal infections, as well as to prevent their recurrence and re-establish the homeostasis of vaginal microbiota.

The microstructure of our hydrogels relies on purified never-dried bacterial nanocellulose (NDBNC), which ensures hydrophilicity and flexibility through its nanofibrils, combined with amphiphilic Poloxamer 407 as binder between the hydrophobic and hydrophilic biocompounds, and with additional chitosan in H3 to increase the hydrogel stability and the mucoadhesion to the vaginal epithelium.

NDBNC is a hydrophilic nano-microfibrillar system in which the nano-dimension is represented by the fiber diameter of 20–30 nm, and the fiber length remains in the range of 1–10 µm after 10–20 passes in a microfluidizer at high pressures, as we previously showed [26]. The low susceptibility to chemical functionalization makes NDBNC more attractive to physical modification and blending than other types of (nano)cellulose; therefore, we aimed to exploit the hydrophilicity and nanofibrillar mesh flexibility of NDBNC.

Poloxamer 407 (Pluronic 188) has the hydrophile-lipophile balance (HLB) of 22, and is predominantly hydrophobic. The gelation of aqueous poloxamer mixtures is closely related to dehydration of the hydrophobic poly(propyleneoxide) blocks, followed by formation of micelles. Additionally, PX increases the sol–gel transition temperature T_Gel_; therefore, it can be proportionally added to obtain specific T_Gel_ materials [114]. The base B1,2 presented in this work consisted of a BNC-PX system of 0.4% BNC and 15% PX, respectively, and the base B3 consisted of 0.4% BNC, 5% PX, and 3% chitosan (CS). The B3 base was synthesized similarly with the H3 synthesis but in the absence of extracts and additives, namely, by spiking the BNC0.4-PX5 suspension with 1 mL acetic acid per 100 mL BNC-PX solution, followed by the addition of powdered CS and homogenization. Via this procedure, the interaction between BNC-PX-acetic acid and CS should be increased, where acetic acid acts like a ligand with one end bound to BNC by hydrogen bonds between its hydroxyl group and BNC hydroxyls in the pre-mixed BNC-PX system, and with the carboxylic end bound to chitosan by ionic interactions. This fact is suggested by the homogeneous aspect of the B3 and H3 and particular analytic fingerprints, as further discussed.

The TEM micrographs showed that PX forms micelles that have the tendency to align along the BNC nanofibrils. PX micelles of around 10–20 nm were previously observed by TEM and DLS [75,115], and micelles with a mean diameter of 60 nm were evaluated for a poloxamer P403-heparin system [116]. The micelles in Figure 2d appear to have the same size as the diameter of some of the bacterial cellulose nanofibrils, which were evaluated in our previous study of BNC to be around 10–20 nm [26], or around 40–60 nm in another study [29], which could favor their interactions. The variation in micelle diameter might be related to the incorporated amount of biocompounds, especially hydrophobic compounds such as thyme essential oil.

FTIR spectroscopy showed the bands of H1 and H2 to be dominated by PX with its particular C-C-O and -CH_3_ absorption bands. The BNC influence, the PX-BNC interaction, and the contribution of the other compounds are evidenced by small bands and shifts in the PX absorption bands in the C-H region around 2882 cm^−1^ and in the polysaccharides region around 1097 cm^−1^. The rheological analysis and X-ray diffractograms were also dominated by PX signals in the binary hydrogels H1 and H2.

The characteristic FTIR bands of CS in solution of acetic acid (CSAcAc) are distinctively observed in H3, being dominant in the corresponding base B3. The interactions between the three polymers were evidenced by small shifts in frequencies from 1543 cm^−1^ and 1024 cm^−1^ in CSAcAc to 1547 cm^−1^ and 1030 cm^−1^ in B3. The dominance of CS over PX is probably related to both the reduced PX concentration and to the coating property of chitosan. The interaction between BNC and CS in a wound-dressing system was previously evidenced to involve the amide bands of CS, namely, amide I around 1613 cm^−1^, amide II around 1550 cm^−1^, and amide III around 1377 cm^−1^ [117]. Conclusively, FTIR spectroscopy evidenced characteristic bands of individual components and their non-covalent and ionic interactions in the complex hydrogels. The homogeneity of hydrogels is depicted in FTIR spectroscopy by transformed absorption bands or bands shifting at different wavenumbers.

The interaction between PX and BNC is also suggested by XRD, i.e., by the shifts in the PX diffraction peaks. In the case of XRD, probably the most interesting diffraction aspect is the unchanged peak at 12.12°; this might be related to a citrate salt, which is also correlated with the FTIR band around 1730 cm^−1^, or the ionic system in which citric acid is involved. Citric acid, present in the same amount of 3% in all hydrogels, is a highly crystalline compound with many peaks, but without particular diffraction peaks in hydrogels. Another interesting peak is that at approx. 21.8°, which is shifted from 22.7° in pure BNC. This shift suggests higher predominance of Iα and/or conformational changes of BNC within bases and hydrogels compared to pure BNC. This would be the first report suggesting these types of changes in cellulose packaging, which could be induced by physical bond formation between BNC and the other components.

The XRD data also showed strong interaction between polymers in B3 and H3, leading to a significantly decreased crystallinity, a decreased intensity of all peaks, and an absence of chitosan peaks. This strong interaction is also suggested by the rheology data, as H3 lacks a gel transition temperature, probably because of the lower PX concentration and the influence of chitosan.

The hydrogels H1 and H2 showed a sol–gel transition temperature of around 18 °C for H1 and 22 °C for H2. Thermogelling systems have been developed to improve vaginal drug delivery due to their liquid form at room temperature, followed by sol–gel transition at physiological temperature. Additionally, after gelling, these systems usually exhibit mucoadhesion to improve retention in the vaginal cavity and the delivery time of bioactives. The gelling temperature (T_gel_) specific to each thermogelling system is a crucial parameter for their performance and ranges between 15 and 37 °C. Values closer to the physiological temperature are most desirable for an efficient application.

The thermogelling properties of the vaginal formulations rely on the use of some specific polymers, among which poloxamers were the most studied due to their biocompatibility and temperature-dependent hydrophilic–hydrophobic properties. At physiological temperature in suitable interactions, poloxamer solutions undergo a change in micellar properties and hydrophobic interactions that leads to a reversible sol–gel transition [71]. In our case, H1 and H2 should be kept at refrigerating temperatures (3–5 °C) during long storage to maintain a liquid state and to preserve the biocompounds, and be applied at room temperature, since the viscosity and the stiffness start to increase around 18 °C for H1 and around 22 °C for H2, and have a good flowing behavior at room temperature and become soft gels at body temperature. Although not particularly analyzed, the first batches dating 7 months ago still show a good homogeneity and consistency in refrigerated storage conditions, but this aspect will be further investigated. Hydrogels based on poloxamer for metronidazole and curcumin delivery had gel transition temperatures of between 29 and 33 °C, and an increase in viscosity with temperature from 100 Pa·s at 25 °C to 500–3000 Pa·s at 37 °C [118]. The viscosities of our hydrogels are lower, namely, around 2 Pa·s for H1 and H2, and around 30 Pa·s for H3. Moreover, the stress values of around 14 Pa for H1 and H2, and 300 Pa for H3, are comparably lower than those for a nanocellulose-gellan cross-linked scaffold for the vaginal delivery of fluconazole, which are around 1–2 MPa [32]. Our hydrogels have a viscosity close to but a little higher than that of a BNC-PX system containing 18.5–22% PX intended for prolonged release of octenidine to treat skin wounds [73]; their system viscosity ranges from 0.1 Pa·s at 10 °C to 0.9 Pa·s at 32 °C. It can be concluded that our hydrogels have average viscosities compared with those of other reported formulations.

The oscillatory sweep experiments were performed between 0.1 and 100 rad/s, while the flow sweep tests were performed between 0.1 and 100 s^−1^. According to the generalized flow curve, the shear rates from 0.1 to 1 s^−1^ correspond to flow draining under gravity or sedimentation during storage, and the shear rates between 1 and 1000 s^−1^ correspond to pipe flowing, with the particular ranges of 10–100 s^−1^ for dip coating and of 10–1000 s^−1^ for mixing and steering [98]. Particular physiological processes related to the vaginal area are estimated between 0.1 and 100 s^−1^ shear rates [101]. Regarding the topical application of the hydrogels with a dispenser, the high frequencies or shear rates from 10 to 100 s^−1^ can be related to shear forces during hydrogel application by pump pressing of a dispenser or an airless vacuum pump, while the low frequencies from 0.1 to 10 s^−1^ can be related to gravity leveling or slow internal spreading of the hydrogel upon skin contact. From this practical application point of view, the UP mode in Figure 4 corresponds to hydrogel shearing behavior during pushing the dispenser, while the reverse R mode corresponds to hydrogel relaxation behavior after shearing. Our formulations worked well with a commercial airless vacuum pump jar with a diaphragm on a spring, which restricts air contact of the product and protects the sensitive compounds, in our case the phenolic compounds and the volatiles, from oxidation or contamination.

By comparison with common fluids such as cosmetic creams or body lotions, our hydrogels have the consistency of a body cream, especially H3, with G′ and G″ values around 100 Pa and initial complex viscosity between 500 and 1000 Pa·s [98]. H1 and H2 are more like diluted body creams.

The phase angle variations for the three hydrogels can be interpreted regarding their practical application in the following way. The hydrogels are in steady state at room temperature, like creamy liquids, with the viscous (liquid-like) behavior in the order of H3 > H1 > H2 suggested by the initial values of the phase angle δ. With the increase in the angular frequency ω during pumping, the phase angle decreases, and the elastic (solid-like) behavior appears, with the minimum δ values remaining in the flowing range between 13° for H2 and 20° for H1 and 23° for H3. At high ω values between 5 and 100 rad/s, the δ trend for H1 remains relatively constant, δ of H2 increases and H2 becomes more fluid, and δ of H3 constantly decreases and H3 becomes thicker. The minimum values of δ for H1 and H2 also correlate with an existing yield stress, which was later confirmed and estimated with the Hershel–Bulkley model. Additionally, the differences in δ variation might be later correlated with the individual customer’s perception of creaminess, softness, tackiness, or firmness in a feedback evaluation panel.

Chitosan is a well-known ionic biopolymer, and its properties were exploited in formulating mucoadhesive hydrogels with various applications, including vaginal hydrogels [66,119,120]. In our H3 hydrogel containing 0.4% BNC, 5% PX, and 3% CS, chitosan strongly influenced the FTIR with its N^+^COO^−^ ionic groups, and also overwhelmed the strong XRD signal of PX, while in the rheological experiments it showed a stabilizing effect on temperature and shear rate variations. Based on FTIR, XRD, rheology, and TEM images, we describe the ternary BNC-PX-CS hydrogel as a flexible, stable, and mucoadhesive system with 25 nm PX micelles arranged as nano-pearls on the nanofibrillar chains of bacterial cellulose coated with chitosan. The chitosan coating is observed in TEM micrographs as a shadow surrounding the bacterial cellulose nanofibrils with a relatively constant layer thickness, and double the nanofibril diameter, i.e., around 50 nm. Mucoadhesion is a time-dependent contact phenomenon induced by a multitude of weak, non-covalent forces such as hydrogen bonds, Van der Waals forces, and ionic interactions. From a biological perspective, mucoadhesion is an important phenomenon in order to maintain the integrity of the vaginal mucosa, which is the physiological and immunological barrier against pathogens. Due to the pattern recognition receptors (PRRs), which are located on the surface and bind to pathogen-associated molecular patterns (PAMPs), the vaginal mucosa represents a real line of defense triggering immunological responses [121]. The surface of the mucosa (mucosal surface layer) contains mucins (highly glycosylated proteins containing sialic acid and sialoglyco proteins), and is a dense but also well-lubricated area [122]. In bacterial vaginitis infections, *Gardnerella vaginalis* grows out of control and produces sialidase, an enzyme capable of cleaving mucins to the remaining sialic acid, and disrupts the membrane integrity [111]. Once the physiological barrier is weakened, the infections can easily spread to the genital tract and severe complications can occur.

Our results indicated a mucin binding efficiency of about 60% in the case of B1,2 and B3 at 44.8 base/mucin ratio (mg/mg), which suggests a high potential to contact mucins from the mucosal surface layer and exhibit their biological activity at the target site.

H1, H2, and H3 showed similar trends in cell viability and proliferation. At the intermediate concentrations tested of 25 and 50 µg/mL, H2 produced a marginally significant increase in the number of metabolically active cells compared to H1. The only difference between these two hydrogels is the addition of *Opuntia ficus-indica* powder in H2. H3, on the other hand, at the concentration of 50 µg/mL, was able to increase the number of metabolically active cells more than H1 and H2, suggesting that the presence of chitosan and/or the reduction of some components is beneficial in this regard. Depending on the source and degree of deacetylation, chitosan is able to support cell growth and adhesion [123], showing a great potential for wound healing [41,42,43,44]. CS acts together with NDBNC, which showed biocompatibility and supported the adhesion and growth of human dermal fibroblasts on BNC hydrogel in previous studies [124].

Stimulating the growth of lactobacilli is an extremely important aspect when it comes to vaginal infections, with lactobacilli being a key component for defense against pathogens [56,125,126]. In our case, H2 promoted the growth of *L. reuteri* 48 h and 72 h after the treatment, with the highest tested concentrations suggesting potential for equilibrating the vaginal microbiota homeostasis. At 48 h after the treatment with the highest tested concentration of H3, a marginally significant increase in *L. reuteri* growth was also observed, which highlights a possible higher efficiency at concentrations higher than those tested.

The antimicrobial activity is another important aspect of the formulations intended for vaginal infections. *E. coli* is part of group B streptococci (GBS), which are the main triggers of aerobic vaginitis [127]. In approximately 20% of the cases, it was demonstrated that aerobic vaginitis coexists with bacterial vaginosis, which leads to an increased risk of severe complications in the absence of treatment [70]. *C. albicans*, on the other hand, is the main cause of VVC (in approximately 90% of cases), with the main risk factors being antibiotics, oral contraceptives that contain a high amount of estrogen, sexual activity, and inhibitors that lead to a decrease in the blood sugar level of patients with type 2 diabetes [121,128]. In terms of antimicrobial activity determined by the diffusimetric method, all hydrogels were active against *E. coli* and *C. albicans*. The antibacterial effect of H3 was also due to acetic acid, which has been shown to be effective against many bacterial strains [129]. By investigating the antimicrobial activity at different microbial densities of the 50 µg/mL hydrogel suspensions, which had both the potential to stimulate cell proliferation and to promote lactobacilli growth, H2 exhibited the highest potential in inhibiting *E. coli* 12 h after the treatment, of between 10 and 90%, depending on the bacterial cell density. However, we emphasize that the 90% inhibition was only for the lowest bacterial density of 1.5 × 10^1^, while, 24 h after treatment, the percentage of inhibition decreased, and H2 was less effective. This indicates that the hydrogels should be further optimized. H2 and H3 were equally effective in inhibiting *C. albicans*. H2 completely inhibited *E. coli* biofilm formation and H1 inhibited about 60% of bacterial biofilm at 1.5 × 10^7^ and 1.5 × 10^6^ bacterial cells. At lower bacterial densities, H1 inhibited the biofilm formation completely. H3 reduced biofilm formation by approximately 23–37%, depending on the bacterial density. The results of this assay, in which the microbial density was varied, indicate a promising capacity for prevention of microbial growth and biofilm formation. This is extremely necessary in the case of recurrent vulvovaginal candidiasis (RVVC) infections in which there are more than three VVC infections per year. RVVC affects approximately 8% of women globally (about 140 million cases per year) and its risk factors are less known, with recent studies pointing to a genetic susceptibility [128].

Our hypothesis for the ternary NDBNC-PX-CS system studied in this work was that NDBNC will ensure flexibility and stability through its nanofibrillar network, together with high hydrophilicity due to its never-dried state; amphiphilic PX will entrap hydrophobic biocompounds such as essential oil molecules, while sticking with the hydrophilic marginal chains to NDBNC; and the third polymer, CS, will increase the stability of the hydrogel via cohesion of the NDBNC-PX system and the mucoadhesion through hydrogen bonding. While confirming most of these assumptions, the experiments showed that the most suitable properties and the highest bioactivity was obtained for H2 instead of H3. It is possible that overly strong interactions and excessive stability are detrimental to the properties needed.

## 4. Materials and Methods

### 4.1. Materials

For the preparation of the hydrogel systems, never-dried bacterial nanocellulose (NDBNC) was used, obtained by alkaline purification from Kombucha membranes and a cascade of mechanical treatments ending with microfluidization, as previously described [26]. BNC was mixed with Poloxamer 407 having the approximate molecular formula PEG_101_PPG_56_PEG_101_ and average molecular weight 12.6 kDa (Sigma-Aldrich, St. Louis, MO, USA) and with high molecular weight chitosan 190–375 kDa practical grade, deacetylation degree > 75% (Sigma Aldrich, St. Louis, MO, USA) after the procedure as further described.

The biocompatibility assay was performed using NCTC clone 929 (L cell, L-929, derivative of Strain L) CCL-1 purchased from ECACC (Sigma-Aldrich, Darmstadt, Germany). For probiotic growth assessment, the following lactobacilli strains were used: *Limosilactobacillus reuteri* DSM 20016 and *Lactiplantibacillus plantarum* subsp. *plantarum* DSM 1055. In order to evaluate the antimicrobial activity, the following microbial strains were used: *Escherichia coli* ATCC 25922 and *Candida albicans* ATCC 10231. The following chemicals were used: Minimum Essential Medium (MEM), fetal calf serum (FCS), L-glutamine, trypsin (E.C. 3.4.21.4), ethylenediamine tetraaceticacid (EDTA), 3-(4,5-dimethylthiazol-2-yl)-2,5-diphenyltetrazolium bromide (MTT), chitosan, lactic acid, citric acid, mucin from porcine stomach type II, antibiotic antimycotic solution 100× (Sigma-Aldrich, St. Louis, MO, USA), Müeller-Hinton agar, Sabouraud dextrose agar, MRS Agar, Müeller-Hinton broth, Sabouraud dextrose broth, MRS broth, potassium iodide, fuchsin basic for microscopy, crystal violet (Scharlau, Barcelona, Spain), glacial acetic acid, potassium metabisulfite, sodium chloride, hydrochloric acid, activated charcoal (Chimreactiv, Bucharest, Romania), periodic acid (VWR International, Radnor, PA, USA), methanol (Honeywell, Wabash, IN, USA). Pure inulin was obtained from CaliVita International (Bucharest, Romania), hydro-glycero-alcoholic extract of *Vitis vinifera* seed was purchased from PlantExtrakt (Cluj, Romania), thyme (*Thymus vulgaris*) essential oil was obtained from Mayam (Oradea, Romania), and powder of *Opuntia ficus-indica* from Laboratoarele Medica, Romania.

### 4.2. Preparation of Hydrogels

After a number of dissolution and homogenization tests, the best procedure was chosen as further detailed (Table 3). The hydrophilic phase was represented by the BNC nano-fibrillar suspension in water, 0.4% *w*/*v*, in which the water-soluble compounds (3% *w*/*v* inulin, 3% *w*/*v* citric acid (for H1, H2, H3), and 0.1% *w*/*v* powder of *Opuntia ficus-indica* (for H2)) were dispersed and dissolved using an ultrasonic bath for 30 min. The hydrophobic phase was represented by PX at room temperature with its hydrophobic PPG core, on which the hydrophobic compounds were directly added, namely, 0.5% *v*/*v* thyme essential oil, 0.5% *v*/*v* hydro-glycero-alcoholic extract of *Vitis vinifera*, and 6% *v*/*v* (for H1, H2), namely, 3% *v*/*v* (for H3) lactic acid, all homogenized in a paste. For H3, which contains 3% *w*/*v* CS, the hydrophilic phase previously described was spiked with 1 mL acetic acid per 100 mL BNC solution, followed by the addition of 3 g of CS and ending with ultrasonic homogenization for 30 min. After the homogenization of each phase, the hydrophilic and hydrophobic suspensions were cooled in a refrigerator at 3–5 °C overnight (≈16 h). The next day, the cooled suspensions were mixed and homogenized using an Ultra-Turrax homogenizer (Ultra-Turrax^®^, IKA, Staufen, Germany) at 600–1200 rpm on an ice bath at t < 5 °C, when PX relaxes its PEG chains and becomes more hydrophilic. The final hydrogels were stored in a refrigerator for further analyses. The pH of the hydrogels was assessed using a pH-meter Seven Compact 2S10 (Mettler Toledo, Columbus, OH, USA).

In order to study the interaction with mucin, a suspension of 3.5% mucin in double-distilled water was prepared and left for one hour at room temperature in a Loopster digital rotating shaker (IKA, Staufen, Germany), 80 rpm. The suspension is referred to as Mu. Subsequently, H1, H2, and H3 hydrogels were mixed using a 1:1 ratio with the 3.5% mucin suspension (Mu), resulting in H1 + Mu, H2 + Mu, and H3 + Mu samples. The samples were analyzed in their initial state, except for the analyses where it is mentioned that the samples were freeze-dried. A ScanVac CoolSafe 55-4 freeze-dryer (LaboGene, Bjarkesvej, Denmark) was used for the freeze-drying process, the working temperature being −55 °C.

### 4.3. Hydrogel Ultrastructural Characterization

Scanning electron microscopy (SEM) images of the freeze-dried hydrogels with and without mucin were acquired with TM4000Plus II tabletop electron microscope (Hitachi, Tokyo, Japan) working with 15 kV electron acceleration voltage, backscattered electrons (BSE) detector, high (H) vacuum mode, 2.5 k magnification.

For acquisition of images in the scale 20–2000 nm, a transmission electron microscope Tecnai™ G2 F20 TWIN Cryo-TEM (2015-FEI Company™, Hillsboro, OR, USA), working at 30 kV in LFD mode, was used. Hydrogel samples were easily prepared by pouring a small droplet of aqueous suspension on a holey carbon grid, without staining, due to a sufficient contrast given by the samples.

### 4.4. FTIR Characterization

Fourier transform infrared (FTIR) spectroscopy was performed on freeze-dried samples in the attenuated total reflectance (ATR) mode on anIRTracer-100 spectrometer (Shimadzu, Kyoto, Japan), in the wavenumber range 4000 to 400 cm^−1^, by accumulation of 45 spectra at 4 cm^−1^ resolution. The exported .txt files were graphically processed using OriginPro2022b software version 9.9.5 from OriginLab Corporation (Northampton, MA, USA).

### 4.5. X-ray Diffraction

The X-ray diffraction (XRD) analyses were performed on freeze-dried samples using a Rigaku diffractometer (Rigaku Corporation, Tokyo, Japan) equipped with SmartLab1.3.3.0 software. The XRD analyses were obtained with an incident Cu_Kα1_ radiation (λ = 1.54059Å) working at 40 kV and 200 mA emission current. The diffractograms were acquired in the range of Bragg’s angles 2θ 5–50° with a resolution step of 0.02° and 4°/min scan speed. The diffraction spectra were smoothed in the PDXL 2.7.2.0. software using the B-Spline model for Chi = 1, followed by optimized deconvolution, crystalline and amorphous peak identification, and calculation of the crystallinity degree (Xc, %) as the ratio between the area of crystalline peaks over the total peaks’ area. The final graphical representations were obtained by processing the exported .csv files with the help of OriginPro2022b software version 9.9.5 from OriginLab Corporation (Northampton, MA, USA).

### 4.6. Viscosity and Mucoadhesion Determination by Rheology

The rheological studies of the hydrogels and their interaction with mucin were performed using a HR20 Discovery Hybrid rotational rheometer from TA Instruments, (New Castle, DE, USA) in three different shearing modes with hysteresis (up and reverse speed variation) at 25 °C, and between 10–40 °C, using a 1000 µm geometry gap. A sample amount of around 1 mL, able to fill the 1000 µm gap between the 40 mm geometry and glass surface, was subjected to oscillation mode in the angular frequency range of ω 0.1–100 rad/s, followed by linear flow sweep in the shear rate range of 1–100 s^−1^, temperature variation for both oscillatory and flow modes, and ending with the axial mode of geometry rising with constant speed of 20 µm/s for a duration of 5 min to determine the adhesion force and adhesion time. The adhesion energy (A_E_) was calculated with Equation (1):A_E_ = F × d/S, (1)
where F is the adhesion force identified as the axial force, d is the work distance interpolated in the axial force graphics, and S is the contact surface with the 40 mm diameter geometry (cylinder), meaning S = 1.257 × 10^−3^ m^2^. The rheological curves were fitted with the available functions in the Trios software version 5.1.1 from TA Instruments. The crossover modulus G′ = G″, where available (just for H3), was determined by cubic spline interpolation using the Trios 5.1.1 software.

### 4.7. Quantitative Mucoadhesion Assay of the Hydrogel Systems

The two hydrogel bases were mixed with mucin suspension using a ratio base/mucin of 44.8 or 15 and incubated at 37 °C for 1 h (Static Cooled incubator MIR-154 PHCbi, St. Louis, MO, USA), under shaking conditions (Trayster IKA, Staufenim Breisgau, Germany). Afterwards, the mixtures were centrifuged for 1 h at 20,000× *g*, at room temperature (Universal 320R Centrifuge Hettich, Tuttlingen, Germany). The periodic acid Schiff’s (PAS) reaction was used for quantifying the free mucin in the supernatant [130,131]. Briefly, after mixing 1 mL of supernatant with 100 μL periodic acid solution prepared by adding 10 µL of 50% periodic acid to 7 mL of 7% acetic acid, the suspensions were incubated for 2 h at 37 °C under shaking. Afterwards, 100 μL Schiff’s reagent was added. The absorbance was measured at λ = 555 nm using a microplate reader (CLARIOstar BMG Labtech, Ortenberg, Germany), after 30 min of incubation in the dark. The calibration curve was performed in the concentration range of 0–0.07% from a mucin stock solution of 0.1%. The mucin binding efficiency was calculated using Equation (2):(2)$$Mucin\;binding\;efficiency\;\left(\%\right)=\frac{Ci-Cf}{Ci}\times 100$$
where *Ci* is the initial concentration of mucin in the reaction mix, and *Cf* is the free mucin in the supernatant.

### 4.8. Cell Viability Assay

The NCTC cell line (clone 929) was grown in T75 flasks in MEM supplemented with 10% FCS, 2 mM L-glutamine, and 1% mixture of antibiotics. The culture was maintained in an incubator with humidified atmosphere, under 5% CO_2_ and at 37 °C. For experiments, the cells were harvested from sub-confluent cultures using 0.25% trypsin-0.53 mM EDTA solution and were re-suspended in fresh serum-supplemented MEM before plating.

The biocompatibility of the hydrogels was assessed based on the mitochondrial succinate dehydrogenases activity (MTT assay), as previously described by Mosmann [132]. Briefly, cell suspension (5 × 10^3^ cells/well) was seeded in 96-well culture plates and was incubated under a 5% CO_2_ humidified atmosphere, at 37 °C, for 24 h. Then, sterile suspensions of hydrogels were added in the culture medium and the plates were incubated at 37 °C, under standard conditions of cultivation. After 24 h, the culture medium was replaced with fresh medium containing MTT solution, in a 10:1 (*v*/*v*) ratio and the plates were incubated at 37 °C, for 3 h. Then, 100 μL of isopropanol was added to each well to dissolve the formazan crystals by gently shaking on a platform, for 15 min. The optical density (OD) was read at 570 nm, using a microplate reader (Berthold Mithras LB 940, Potsdam, Germany). The measured OD is directly proportional to the cell viability and the results were calculated using Equation (3):(3)$$Cell\;viability\;\left(\%\;of\;C\right)=\frac{OD_{570}\;sample}{OD_{570}C}\times 100$$

The cells cultured in complete culture medium served as control (*C*). Three separate experiments were conducted, and the results were expressed as mean ± standard error (SE).

### 4.9. Probiotic Growth Assay

For determining the hydrogel effect on the probiotic growth, 180 µL of sterile hydrogel suspensions prepared in MRS broth was added to 96-well plates. Afterwards, 20 µL of *L. reuteri* or *L. plantarum* suspension at McFarland 0.5 prepared in 0.85% physiological saline solution was added over the hydrogel suspensions in the plates. The optical density was measured at 600 nm using a microplate reader. The lactobacilli incubated in MRS without hydrogel suspensions served as positive control (C+), with respect to which the bacterial growth percent of each sample was calculated with Equation (4):(4)$$Microbial\;growth\;\left(\%\;ofC+\right)=\frac{OD_{600}\;sample}{OD_{600}\;C+}\times 100$$

Each variant was analyzed in triplicate and the results were expressed as mean ± standard error.

### 4.10. Effects on Pathogenic Bacteria

#### 4.10.1. Antimicrobial Activity

Semi-quantitative screening of antimicrobial activity was carried out using an adapted diffusimetric method [133]. Two doses of hydrogel, namely, 25 and 100 µL, were applied directly onto Müeller-Hinton agar (MHA) or Sabouraud dextrose agar (SDA) in 90 mm Petri dishes, previously seeded with a standardized microbial suspension 0.5 McFarland prepared in 0.85% physiological saline solution from fresh cultures (MHA for *E. coli* and SDA for *C. albicans*). The dose of 25 µL hydrogels diluted 2:1 (*v*/*v*) with double-distilled water was also studied. Subsequently, the Petri dishes were incubated at 37 °C for 24 h. The antimicrobial effect of the hydrogels was quantified by measuring the diameter of the halo, reflecting the absence of microbial growth around the spot, using the ImageJ software, version 2023 [134]. Each sample was tested in triplicate and compared to the solvent control, i.e., the mixture of water, glycerol, and ethanol used for H1 and H2, and the mixture of acetic acid, glycerol, and ethanol used for H3.

For the quantitative screening of the antimicrobial activity, we chose a dose that was found to be biocompatible after performing the MTT assay and that also promoted the growth of lactobacilli. Hydrogel solutions of 50 µg/mL were prepared in sterile Muller-Hinton broth (MHB) for *E. coli* and in Sabouraud dextrose broth (SDB) for *C. albicans* and 180 µL of each solution were transferred to 96-well plates. Subsequently, 20 µL of serial microbial cell dilutions (1.5 × 10^8^, 1.5 × 10^7^, 1.5 × 10^6^, 1.5 × 10^5^, 1.5 × 10^4^, 1.5 × 10^3^, 1.5 × 10^2^, 1.5 × 10^1^) prepared in 0.85% physiological saline solution from standardized cultures of McFarland 0.5 was added over the hydrogel suspensions. The optical density values were measured at 600 nm using a microplate reader. The microbial growth in the absence of hydrogel suspension was used as positive control (C+), from which the percent of microbial growth inhibition was calculated with Equation (5):(5)$$Microbial\;growth\;inhibition\;\left(\%\;of\;C+\right)=\left(1-\frac{OD_{600}sample}{OD_{600}C+}\right)\times 100$$

The microbial cell dilution 1.5 × 10^0^ was considered the negative control, as no microbial growth was observed. Each variant was analyzed in triplicate and the results were expressed as mean ± standard error.

#### 4.10.2. Antibiofilm Activity

After 24 h and the OD reading at 600 nm for the determination of the antimicrobial activity, the wells were washed 3 times with 0.85% physiological saline solution to remove all non-adherent bacterial cells. In order to fix the bacterial biofilm, a 5 min. incubation step in methanol was included, after which the plates were drained and allowed to dry for another 5 min. The next step was the biofilm staining with 0.1% crystal violet solution for 15 min, followed by another washing step with 0.85% physiological saline solution. The optical density was measured at 490 nm using a microplate reader, after resuspending the stained biofilm in 33% acetic acid.

### 4.11. Statistical Analysis

Statistical analysis was performed using the IBM SPSS version 26.0.0.0 Software (One-Way ANOVA). Different letters indicate significant differences between samples.

## 5. Conclusions

Two binary hydrogels (H1 and H2) based on never-dried bacterial nanocellulose (NDBNC) as fibrillar nano-mesh, and Poloxamer 407 (PX) as an amphiphilic binder and thermo-responsive polymer, together with one ternary hydrogel (H3) containing BNC, PX, and chitosan (CS), were loaded with phyto-extracts to obtain biocompatible, antimicrobial, and lactic acid bacteria promoting mucoadhesive systems. Such hydrogels loaded with bioactive phyto-extracts are intended to equilibrate vaginal microbiota imbalances. The nanofibrillar hydrogels with bioactive phyto-extracts were structurally and physical-chemically characterized using TEM and SEM microscopy, FTIR spectroscopy, X-ray diffraction, and rheological behavior with shear and temperature. These physicochemical techniques evidenced good stability under dynamic shearing conditions and visually under 7 months of refrigerated storage conditions, mucoadhesive properties, and structural particularities. TEM micrographs evidenced PX micelles of 20 nm arranged on BNC nanofibrils coated with CS. FTIR spectroscopy evidenced the particular spectra of individual compounds and the convolution of signals in the final hydrogels, dominated by PX bands in H1 and H2, and by CS in H3. The internal, cohesion, energies of the binary hydrogels were evaluated as the difference between the adhesion energies of hydrogels alone minus the adhesion energies of hydrogel–mucin systems and were evaluated to be around 0.875, 1.106, and 8.801 J/m^2^ for the three hydrogels H1, H2, and H3, respectively. Thus, a very high cohesion energy was found in the ternary hydrogel, which correlates with the highest stability. The hydrogels exhibited antimicrobial and antibiofilm activities and a high degree of biocompatibility. The work also demonstrates the potential of our formulation to support cell proliferation and promote the growth of lactic acid bacteria. Overall, the most promising formulation is H2, which is able to have positive results on all aspects, i.e., L929 cell proliferation, prebiotic effect, and antimicrobial activity. Moreover, it has the optimal sol–gel transition temperature among the three hydrogels. However, more optimization is needed in order to improve its bioactivity, for example, its antibacterial effect against *E. coli*.

## Figures and Tables

**Figure 1 pharmaceuticals-16-01671-f001:**
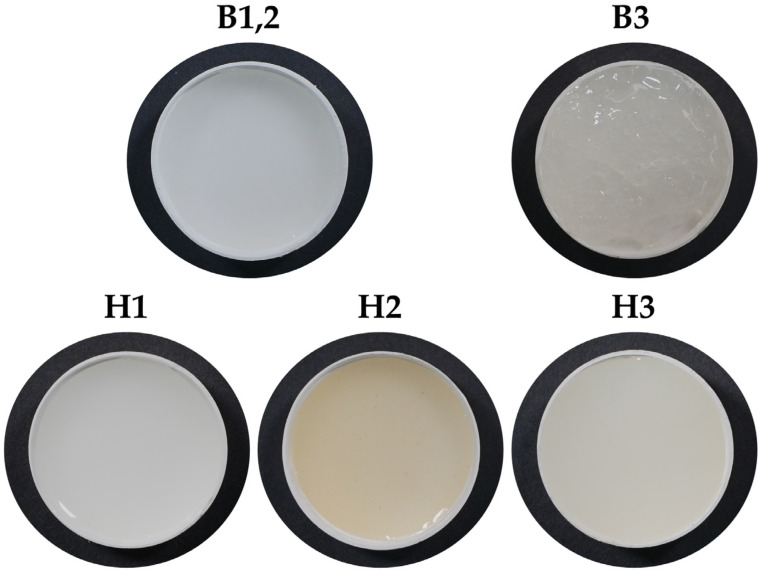
Macroscopic images of bases and hydrogels: B1,2 base for hydrogels H1 and H2 containing BNC 0.4% and PX 15%; B3 base for H3 hydrogel containing BNC 0.4%, PX 5%, and CS 3%; final hydrogels H1, H2, and H3 with the corresponding biocompounds.

**Figure 2 pharmaceuticals-16-01671-f002:**
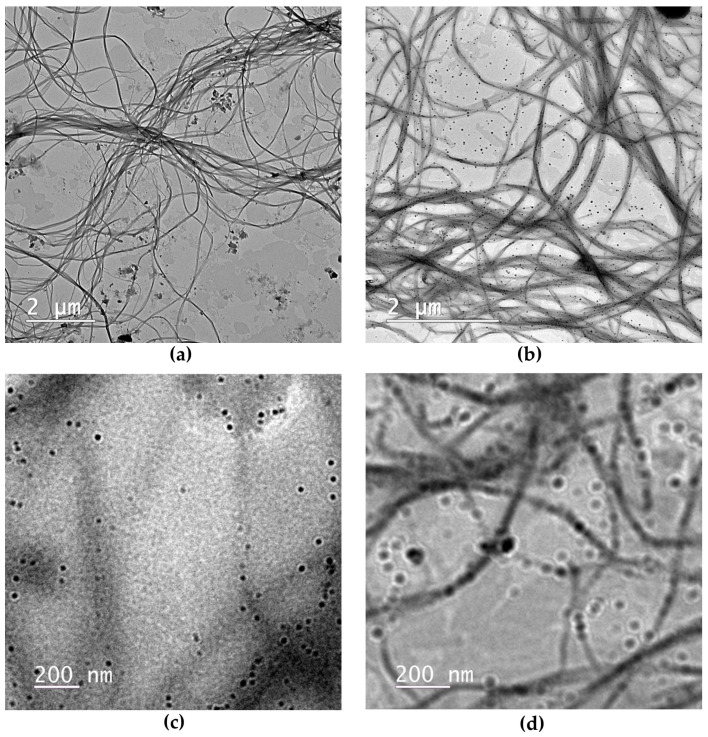
TEM microscopy of hydrogels: (**a**) hydrogel H1 based on BNC 0.4% and PX 15% at microscale; (**b**) hydrogel H3 based on BNC 0.4%, PX 5%, and CS 3% at microscale; (**c**) hydrogel H1 based on BNC 0.4% and PX 15% at nanoscale; (**d**) hydrogel H3 based on BNC 0.4%, PX 5%, and CS 3% at nanoscale.

**Figure 3 pharmaceuticals-16-01671-f003:**
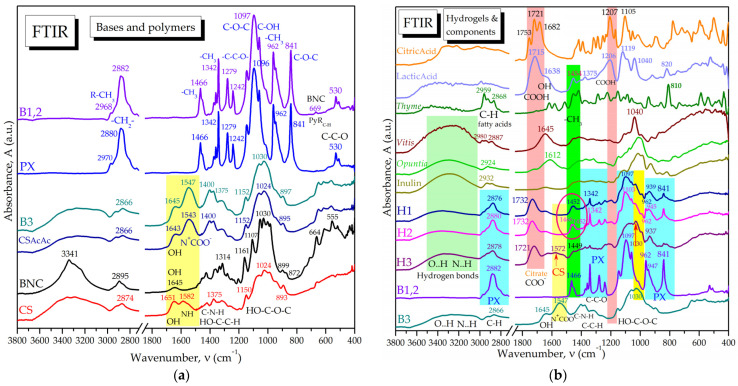
(**a**) FTIR spectra of hydrogel base B1,2 containing 0.4% BNC and 15% PX used as base for the hydrogels H1 and H2, with respect to base B3 containing 0.4% BNC, 5% PX, and 3% CS used as base for the hydrogel H3. The B1,2 and B3 spectra are compared with the individual components PX—poloxamer, BNC—bacterial nanocellulose, CS—chitosan and CSAcAc-3% CS in 1% acetic acid, lyophilized; (**b**) FTIR spectra of hydrogels H1, H2, and H3 in comparison with the two bases B1,2 and B3, respectively, with the components citric acid, lactic acid, thyme essential oil, Vitis vinifera seeds extract, Opuntia ficus-indica powder, and inulin.

**Figure 4 pharmaceuticals-16-01671-f004:**
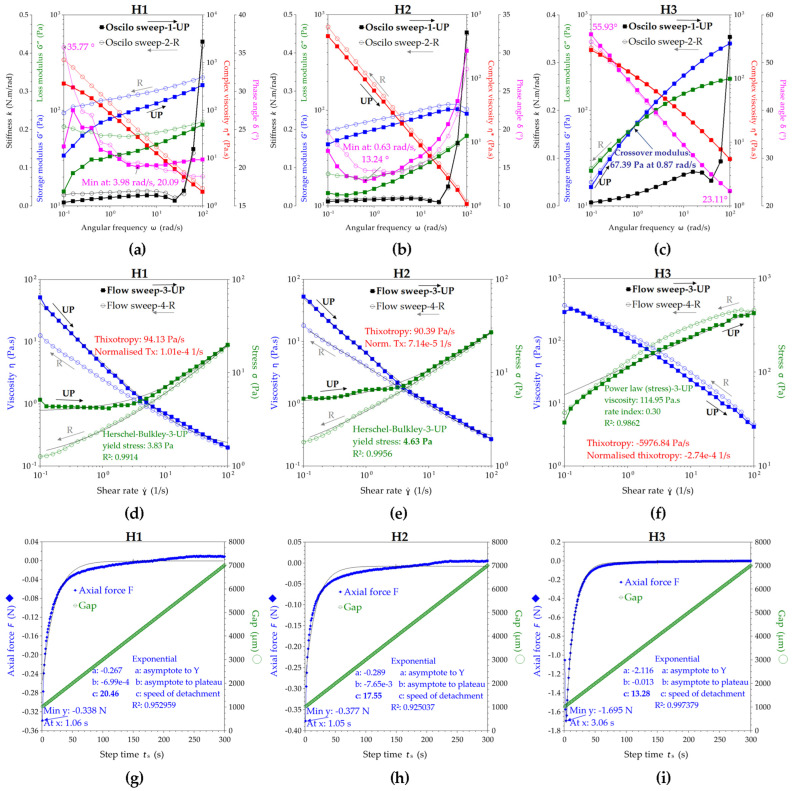
Rheology of hydrogels H1, H2, and H3 in different flow conditions: (**a**–**c**) oscillatory sweep; (**d**–**f**) flow sweep; (**g**–**i**) axial mode. “UP” refers to the increase in the independent parameter on the *x*-axis, while “R” denotes the reverse variation, meaning the decrease in the independent parameter on the *x*-axis, the resulting loops being known as “hysteresis”. The gray lines in (**d**–**i**) represent the regression lines for the mentioned functions.

**Figure 5 pharmaceuticals-16-01671-f005:**
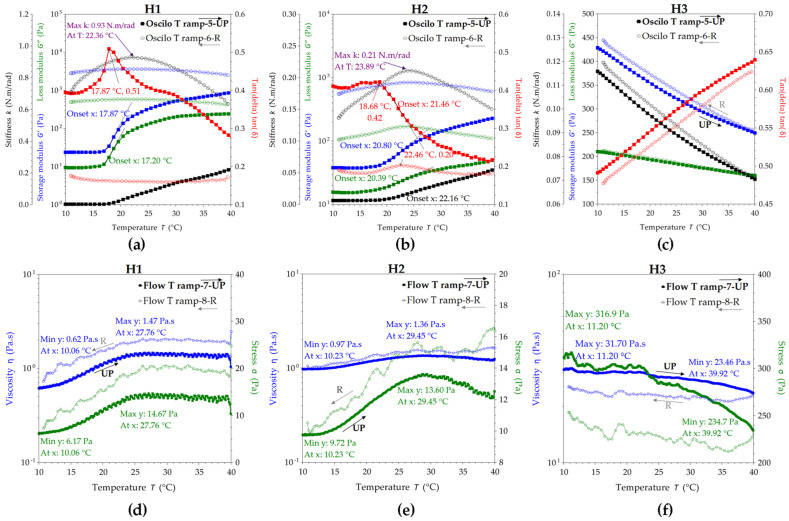
Temperature influence on the rheology of hydrogels H1, H2, and H3 in different flow conditions: (**a**–**c**) oscillatory flow; (**d**–**f**) flow sweep. “UP” refers to the increase of the independent parameter on the *x*-axis, while “R” denotes the reverse variation, meaning the decrease in the independent parameter on the *x*-axis, the resulting loops being known as “hysteresis”.

**Figure 6 pharmaceuticals-16-01671-f006:**
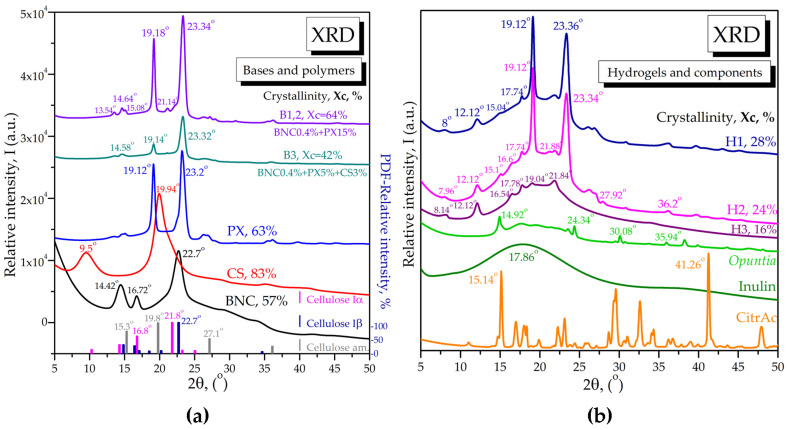
X-ray diffraction analyses of hydrogels: (**a**) XRD of base B1,2 containing 0.4% BNC and 15% PX and of base B3 containing 0.4% BNC, 5% PX, and 3% CS, in comparison with the polymers BNC, CS, and PX; (**b**) XRD of hydrogels H1, H2, and H3 in comparison with the solid components: citric acid—CitrAc, inulin, and *Opuntia ficus-indica*.

**Figure 7 pharmaceuticals-16-01671-f007:**
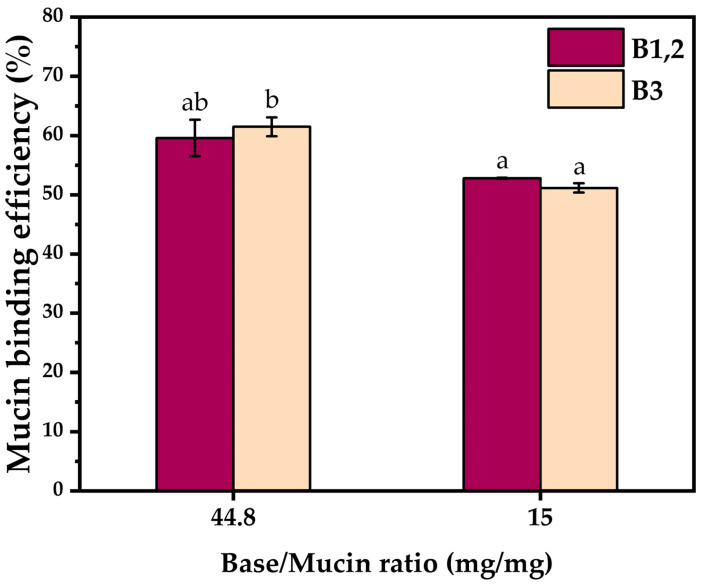
Mucin binding efficiency; B1,2—base of H1 and H2, B3—base of H3 (±error bars, α = 0.05, n = 3, different letters show statistically significant differences between the samples).

**Figure 8 pharmaceuticals-16-01671-f008:**
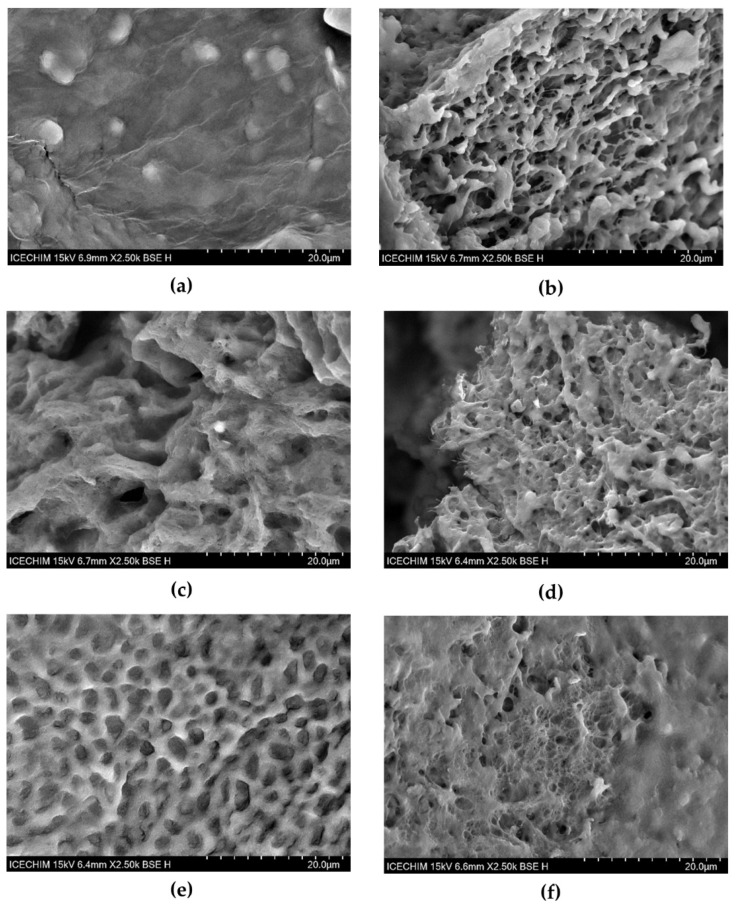
SEM microscopy of hydrogels and their interaction with mucin: (**a**) H1; (**b**) H1 + Mu; (**c**) H2; (**d**) H2 + Mu; (**e**) H3; (**f**) H3 + Mu; H1—Hydrogel 1, H2—Hydrogel 2, H3—Hydrogel 3; H1 + Mu—Hydrogel 1 + 3.5% mucin (1:1, *v*:*v*); H2 + Mu—Hydrogel2 + 3.5% mucin (1:1, *v*:*v*); H3 + Mu—Hydrogel 3 + 3.5% mucin (1:1, *v*:*v*).

**Figure 9 pharmaceuticals-16-01671-f009:**
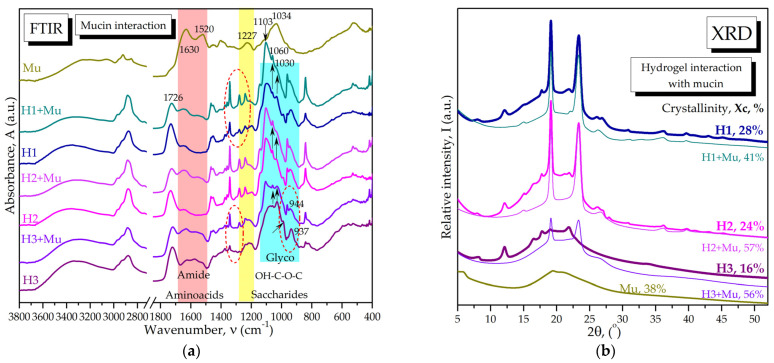
Hydrogels’ interaction with mucin investigated via (**a**) FTIR spectroscopy of hydrogels, and (**b**) X-ray diffraction, where Mu—mucin, H1—Hydrogel 1, H2—Hydrogel 2, H3—Hydrogel 3; H1 + Mu—Hydrogel 1 + 3.5% mucin (1:1, *v*:*v*); H2 + Mu—Hydrogel 2 + 3.5% mucin (1:1, *v*:*v*); H3 + Mu—Hydrogel 3 + 3.5% mucin (1:1, *v*:*v*). The arrows show the reduction by convolution or intensification of the glycosidic bands in hydrogels after the contact with mucin. The red circles highlight other changes in the hydrogels when mixed with mucin.

**Figure 10 pharmaceuticals-16-01671-f010:**
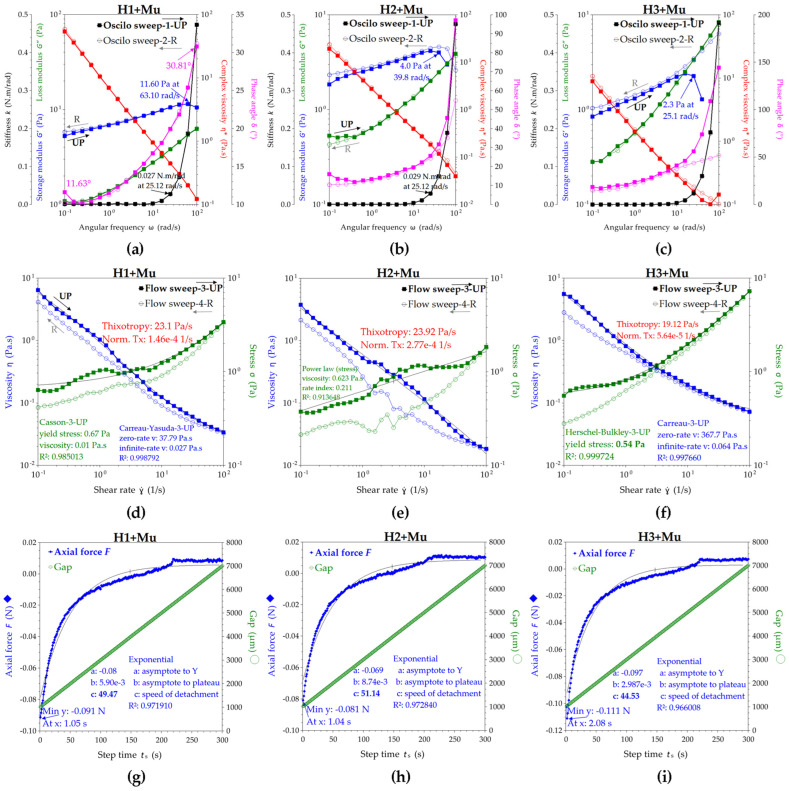
Rheology of hydrogels H1, H2, and H3 in contact with 3.5% mucin aqueous solution in different flow conditions: (**a**–**c**) oscillatory H1 + Mu, H2 + Mu, H3 + Mu; (**d**–**f**) flow sweep H1 + Mu, H2 + Mu, H3 + Mu; (**g**–**i**) axial mode for H1 + Mu, H2 + Mu, H3 + Mu. “UP” refers to the increase in shear, while “R” denotes the reverse variation, meaning the decrease in shear, the resulting loops being known as “hysteresis”. The gray lines in (**d**–**i**) represent the regression lines for the mentioned functions.

**Figure 11 pharmaceuticals-16-01671-f011:**
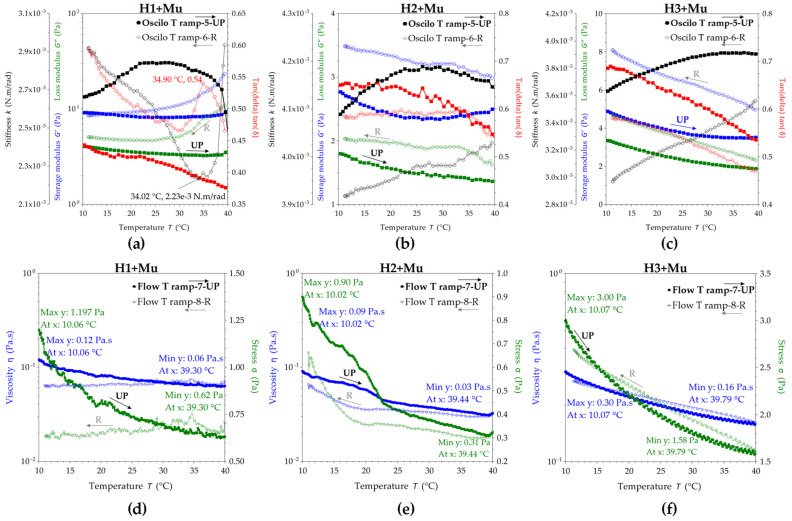
Temperature influence on the rheology of hydrogels with mucin H1 + Mu, H2 + Mu, and H3 + Mu in different flow conditions: (**a**–**c**) oscillatory flow; (**d**–**f**) flow sweep. “UP” refers to the increase in temperature, while “R” denotes the reverse variation, meaning the decrease in temperature, the resulting loops being known as “hysteresis”.

**Figure 12 pharmaceuticals-16-01671-f012:**
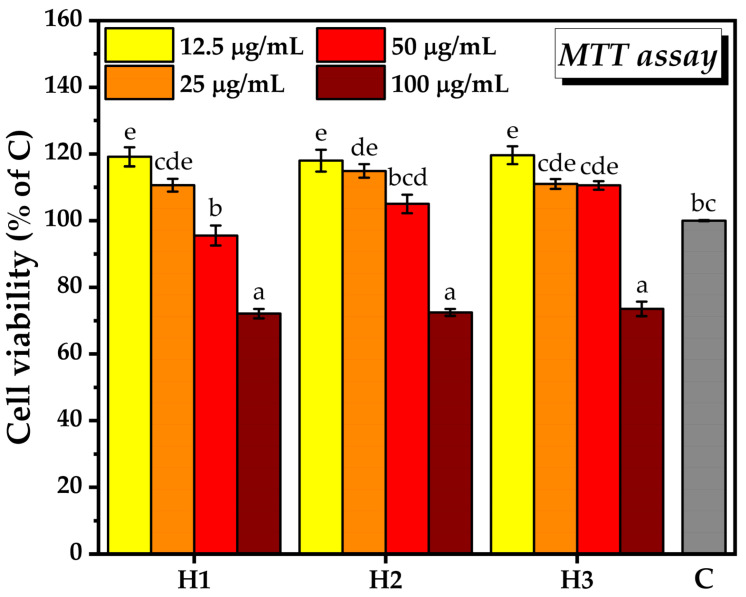
Biocompatibility of the hydrogel systems; H1—cells treated with hydrogel 1, H2—cells treated with hydrogel 2, H3—cells treated with hydrogel 3, C—control, untreated cells; (±error bars, α = 0.05, n = 3, different letters show statistically significant differences between the samples).

**Figure 13 pharmaceuticals-16-01671-f013:**
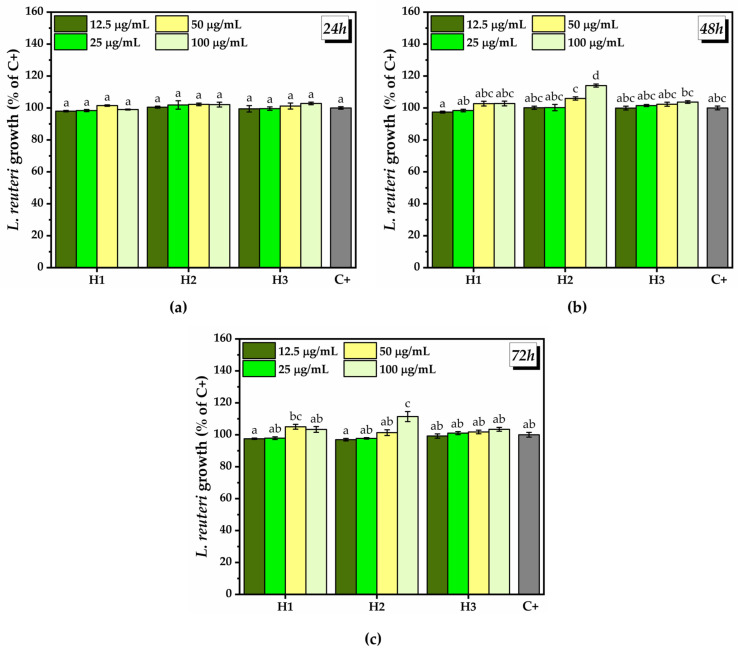
*L. reuteri* growth: (**a**) *L. reuteri* growth after 24 h incubation; (**b**) *L. reuteri* growth after 48 h incubation; (**c**) *L. reuteri* growth after 72 h incubation; H1—Hydrogel 1, H2—Hydrogel 2, H3—Hydrogel 3, C+—Positive control (±error bars, α = 0.05, n = 3, different letters show statistically significant differences between the samples).

**Figure 14 pharmaceuticals-16-01671-f014:**
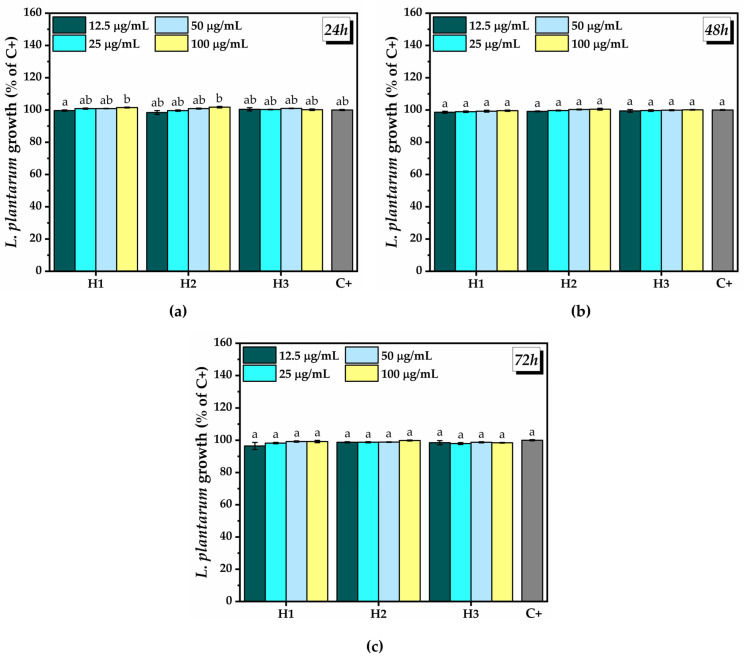
*L. plantarum* growth: (**a**) *L. plantarum* growth after 24 h incubation; (**b**) *L. plantarum* growth after 48 h incubation; (**c**) *L. plantarum* growth after 72 h incubation; H1—Hydrogel 1, H2—Hydrogel 2, H3—Hydrogel 3, C+—Positive control (±error bars, α = 0.05, n = 3, different letters show statistically significant differences between the samples).

**Figure 15 pharmaceuticals-16-01671-f015:**
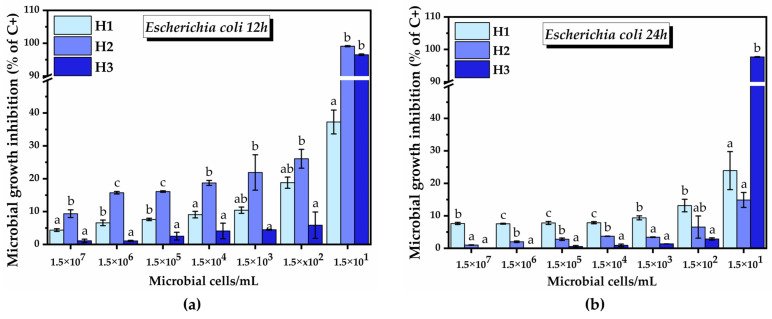
Quantitative screening of antibacterial activity: (**a**) *E. coli* growth inhibition 12 h after hydrogel treatment; (**b**) *E. coli* growth inhibition 24 h after hydrogel treatment; H1—Hydrogel 1, H2—Hydrogel 2, H3—Hydrogel 3 (±error bars, α = 0.05, n = 3, different letters show statistically significant differences between the samples).

**Figure 16 pharmaceuticals-16-01671-f016:**
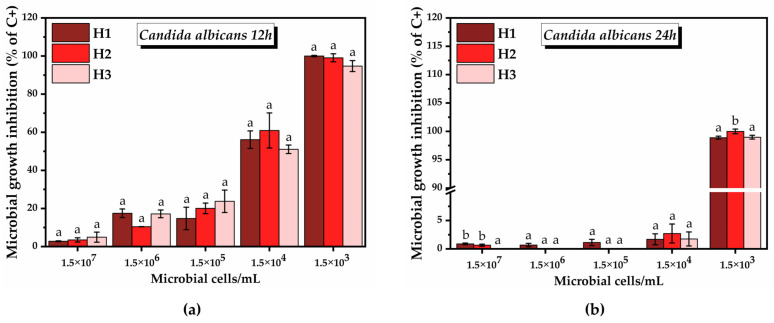
Quantitative screening of antimicrobial activity: (**a**) *C. albicans* growth inhibition after 12 h of incubation; (**b**) *C. albicans* growth inhibition after 24 h of incubation; H1—Hydrogel 1, H2—Hydrogel 2, H3—Hydrogel 3 (±error bars, α = 0.05, n = 3, different letters show statistically significant differences between the samples).

**Table 1 pharmaceuticals-16-01671-t001:** Semi-quantitative screening of antimicrobial activity.

Average Diameter of the Inhibition Zone (cm) ± Standard Error (SE)							
Strain	Dose	H1	C1	H2	C2	H3	C3
*E. coli*	25 µL hydrogel	1.13 ± 0.08 c	0	1.01 ± 0.004 b	0	0.74 ± 0.008 a	0.19 ± 0.08
	100 µL hydrogel	1.62 ± 0.01 a	0	1.61± 0.004 a	0	1.74 ± 0.006 b	1.26 ± 0.03
	25 µL of 2:1 diluted hydrogel (*v*/*v*)	1.27 ± 0.01 b	-	1.21 ± 0.02 b	-	0.81 ± 0.05 a	-
*C. albicans*	25 µL hydrogel	0.28 ± 0.01 a	0	0.31 ± 0.01 a	0	1.01 ± 0.01 b	0
	100 µL hydrogel	1.27 ± 0.01 a	0	1.15 ± 0.02 b	0	1.4 ± 0.04 c	0
	25 µL of 2:1 diluted hydrogel (*v*/*v*)	0 a	-	0 a	-	0.65 ± 0.01 b	-

H1—hydrogel 1, H2—hydrogel 2; H3—hydrogel 3; C1—Solvent control of H1; C2—Solvent control of H2, C3—Solvent control of H3; (α = 0.05, n = 3, different letters show statistically significant differences between the samples).

**Table 2 pharmaceuticals-16-01671-t002:** Antibiofilm activity of the hydrogels for *E. coli*.

Inhibition of Bacterial Biofilm (% of C+)			
	H1	H2	H3
1.5 × 10^7^ bacterial cells	60.75 ± 1.08	100	23.90 ± 3.80
1.5 × 10^6^ bacterial cells	60.40 ± 3.08	100	26.90 ± 1.66
1.5 × 10^5^ bacterial cells	100	100	26.36 ± 1.60
1.5 × 10^4^ bacterial cells	100	100	32.30 ± 0.44
1.5 × 10^3^ bacterial cells	100	100	37.75 ± 2.40

**Table 3 pharmaceuticals-16-01671-t003:** The composition of hydrogels and hydrogel bases.

Compounds	B1,2	H1	H2	B3	H3
PX (*w*/*v*)	15%	15%	15%	5%	5%
BNC (*w*/*v*)	0.4%	0.4%	0.4%	0.4%	0.4%
CS (*w*/*v*)	-	-	-	3%	3%
Inulin (*w*/*v*)	-	3%	3%	-	3%
Thyme essential oil (*v*/*v*)	-	0.5%	0.5%	-	0.5%
Hydro-glycero-alcoholic extract of *Vitis vinifera* (*v*/*v*)	-	0.5%	0.5%	-	0.5%
*Opuntia ficus-indica* powder (*w*/*v*)	-	-	0.1%	-	-
Lactic acid (*v*/*v*)	-	6%	3%	-	3%
Citric acid (*w*/*v*)	-	3%	3%	-	3%

## Data Availability

All the data are presented in this work.

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
