# Peer review of "Bioactive-Loaded Hydrogels Based on Bacterial Nanocellulose, Chitosan, and Poloxamer for Rebalancing Vaginal Microbiota"

_pharmaceuticals, 2023, doi:10.3390/ph16121671_

Round 1

Reviewer 1 Report

Comments and Suggestions for Authors

 The manuscript prepared by  Angela Moraru, Ștefan Ovidiu Dima, Naomi Tritean, Elena Iulia OpriÈ›a, Ana-Maria Prelipcean, Bogdan Trică, Anca Oancea, Ionut Moraru, Diana Constantinescu-Aruxandei, Florin Oancea “Bioactive-loaded ternary hydrogels based on chitosan, nano-cellulose, and poloxamer for rebalancing vaginal microbiota” deals with studies of development of hydrogels loaded with bioactive compounds for equilibrating vaginal microbiota imbalances in order to support the vaginal microbiota homeostasis.

Authors aimed to T to investigate the structural and functional properties of  two Kombucha BNC-PX formulations and a triple-polymeric hydrogel matrix based on Kombucha BNC, CS, and PX loaded with bioactive extracts, in order to support the  vaginal microbiota homeostasis. In my view, this is very important topic as Bacterial vaginosis has been reported in one-third of women worldwide at different life stages.

The manuscript is well prepared, introduction, discussion and conclusions are relevant. The current state of the art is given in introductory part without excessive self-citation. However, still several literature sources dealing with hydrogels for vaginal use, can be included by the authors: Gosecka M, Gosecki M. Antimicrobial Polymer-Based Hydrogels for the Intravaginal Therapies-Engineering Considerations. Pharmaceutics. 2021;13(9):1393. doi: 10.3390/pharmaceutics13091393a or AM dos Santos, SG Carvalho, VHS Araujo, G Corrêa Carvalho, MPD  Gremião, M Chorilli, Recent advances in hydrogels as strategy for drug delivery intended to vaginal infections, International Journal of Pharmaceutics,  590, 2020, 119867, https://doi.org/10.1016/j.ijpharm.2020.119867. Also authors may also mention use of essential oils for treatment of vaginal infections.
Not looking at the fact that there are numerous studies on hydrogels for vaginal use,  to achieve the ultimate goal of improving women’s health and quality of life, a holistic approach to the design of hydrogel-based therapeutic platforms for gynaecological treatment is required (Gosecka M, Gosecki M.).

To the best of my knowledge the bioactive compounds proposed by the authors were not studied before for re-balansing of vaginal microbiota.
The authors for characterisation of elaborated hydrogels use FTIR, TEM, rheology experiments, X-ray, as well as authors studied Hydrogel interaction with mucin, performed biocompatibility assay, prebiotic, antimicrobial activity studies of hydrogels. Do authors consider use of NMR technique or/and Masspectrometry for studies of hydrogels for determination of composition, stability studies or release of active components?

Authors support their studies findings with conclusions, however authors may underline which composition can be more prospective and envisage future perspective of their studies.

Some style points should be corrected before publication: Almost in every chapter, some words are written together without spaces. Perhaps it occurred during the transfer to PDF version. Also, the quality of figures 4, 5, 10, 11, 13-16 can be improved.

The manuscript may attract the interest of readers of ’’Pharmaceuticals’’.
Overall, the manuscript is already in very good shape and can be accepted with minor changes. 

Author Response

Point 1: The manuscript prepared by  Angela Moraru, Ștefan Ovidiu Dima, Naomi Tritean, Elena Iulia OpriÈ›a, Ana-Maria Prelipcean, Bogdan Trică, Anca Oancea, Ionut Moraru, Diana Constantinescu-Aruxandei, Florin Oancea “Bioactive-loaded ternary hydrogels based on chitosan, nano-cellulose, and poloxamer for rebalancing vaginal microbiota” deals with studies of development of hydrogels loaded with bioactive compounds for equilibrating vaginal microbiota imbalances in order to support the vaginal microbiota homeostasis.

Authors aimed to T to investigate the structural and functional properties of two Kombucha BNC-PX formulations and a triple-polymeric hydrogel matrix based on Kombucha BNC, CS, and PX loaded with bioactive extracts, in order to support the vaginal microbiota homeostasis. In my view, this is very important topic as Bacterial vaginosis has been reported in one-third of women worldwide at different life stages.

The manuscript is well prepared, introduction, discussion and conclusions are relevant. The current state of the art is given in introductory part without excessive self-citation. However, still several literature sources dealing with hydrogels for vaginal use, can be included by the authors: Gosecka M, Gosecki M. Antimicrobial Polymer-Based Hydrogels for the Intravaginal Therapies-Engineering Considerations. Pharmaceutics. 2021;13(9):1393. doi: 10.3390/pharmaceutics13091393a or AM dos Santos, SG Carvalho, VHS Araujo, G Corrêa Carvalho, MPD Gremião, M Chorilli, Recent advances in hydrogels as strategy for drug delivery intended to vaginal infections, International Journal of Pharmaceutics, 590, 2020, 119867, https://doi.org/10.1016/j.ijpharm.2020.119867. Also authors may also mention use of essential oils for treatment of vaginal infections.
Not looking at the fact that there are numerous studies on hydrogels for vaginal use, to achieve the ultimate goal of improving women’s health and quality of life, a holistic approach to the design of hydrogel-based therapeutic platforms for gynecological treatment is required (Gosecka M, Gosecki M.).

Response 1: Thank you for your description of our study and your suggestions. We have included the studies in the reference list – line 56.

Point 2: To the best of my knowledge the bioactive compounds proposed by the authors were not studied before for re-balansing of vaginal microbiota.
The authors for characterisation of elaborated hydrogels use FTIR, TEM, rheology experiments, X-ray, as well as authors studied Hydrogel interaction with mucin, performed biocompatibility assay, prebiotic, antimicrobial activity studies of hydrogels. Do authors consider use of NMR technique or/and Masspectrometry for studies of hydrogels for determination of composition, stability studies or release of active components?

Response 2: Indeed, NMR spectroscopy and mass spectrometry are two particular relevant analytical methods for analysis and quantification of biocompounds, but the access to these equipment was not facile this time and we could not perform the additional characterizations in a reasonable timeframe. The FTIR, XRD, TEM, SEM and rheology techniques we used provided significant preliminary Information about these hydrogels and we hope that our results are relevant and accessible to a wider audience. We plan to extend the studies with stability and release of the active compounds, in order to optimize further the hydrogels, but in another study, and we will try to use especially the MS technique which is more sensitive than NMR. The current study is already rather complex, in our opinion.

Point 3: Authors support their studies findings with conclusions, however authors may underline which composition can be more prospective and envisage future perspective of their studies.

Response 3:  We added more information about the most promising formulation in the Conclusions section. 

Point 4: Some style points should be corrected before publication: Almost in every chapter, some words are written together without spaces. Perhaps it occurred during the transfer to PDF version. Also, the quality of figures 4, 5, 10, 11, 13-16 can be improved.

The manuscript may attract the interest of readers of ’’Pharmaceuticals’’.
Overall, the manuscript is already in very good shape and can be accepted with minor changes. 

Response 4: Thank you for your suggestions. Indeed, we also observed that there are some words joined together due to different Word versions we used and we, hopefully, managed to correct all typing mistakes. Also, the figures 4, 5, 10, 11 were improved with bigger font and characters, where possible. In the case of figures 13-16 some colors were rather similar, so we tried to fix this by changing the colors. For figures 13 and 14 we used the multiplication sign from the symbols section. Thank you again for all the suggestions and appreciations on our paper.

Reviewer 2 Report

Comments and Suggestions for Authors

The manuscript entitled "Bioactive-loaded hydrogels based on bacterial nanocellulose, chitosan, and poloxamer for rebalancing vaginal microbiota”. A few comments and suggestions must be addressed before this reviewer recommends the publication of this work in the Journal.

Comments

1.        The author should provide quantitative information in abstract section.

2.        How does the addition of various bioactive compounds, such as inulin and Thyme essential oil, affect the morphological aspect of one of the hydrogels (H3)?

3.        What is the antimicrobial activity of Thyme essential oil (TEO), and how has it been used in formulations for treating vaginal infections?

4.        What are the biological effects of Opuntia ficus-indica extract, and why was it included in the formulations?

5.        Why it is important to combine different concentrations of these compounds in the hydrogel formulations, and what is the overall goal in addressing vaginal infections and microbiota homeostasis?

6.        What is the significance of PX micelles, and how were they observed in the hydrogel structure?

7.        What is the potential role of these hydrogels in supporting cell proliferation and promoting the growth of lactic acid bacteria?

8.        The author provide the outstanding points and highlights of this work in the conclusion section.

9.        Typographical errors and superfluous spaces throughout the manuscript should be corrected.

Author Response

The manuscript entitled "Bioactive-loaded hydrogels based on bacterial nanocellulose, chitosan, and poloxamer for rebalancing vaginal microbiota”. A few comments and suggestions must be addressed before this reviewer recommends the publication of this work in the Journal.

Point 1. The author should provide quantitative information in abstract section.

Response 1: Thank you for suggestions. We tried to provide the main quantitative results in the abstract.

Point 2. How does the addition of various bioactive compounds, such as inulin and Thyme essential oil, affect the morphological aspect of one of the hydrogels (H3)?

Response 2: Inulin and Thyme essential oil are present in the same amounts in all three hydrogels, respectively 3% inulin and 0.5% Thyme essential oil. Particular morphological aspect of H3 is related mainly to chitosan, which was used only for H3, respectively to Opuntia ficus-indica in H2, and we tried to strengthen these aspects in sub-section 2.1 (we added some new Information). On lines 135-136 in the initial manuscript (lines 178-179 now), the pleasant odor of all hydrogels was related to Thyme essential oil. Other particular aspects of individual components were not specifically investigated.          

Point 3. What is the antimicrobial activity of Thyme essential oil (TEO), and how has it been used in formulations for treating vaginal infections?

Response 3: The Information from literature was moved from the Discussion section to the Introduction section, to be easier to follow (lines 96-113). We included additional information in the Discussion section (lines 636-637).

Point 4. What are the biological effects of Opuntia ficus-indica extract, and why was it included in the formulations?

Response 4: We shortened a little the paragraph from the Discussion section where we indicated the main properties of each compound in order to motivate its use in this study.  We moved some Information in the Introduction section, to be easier to follow and understand. Therefore, in the Introduction section now (lines 111-113 we mentioned that “Regarding the biological effect that the extracts from different parts of Opuntia ficus-indica possess, there are several studies that showed its antioxidant, antibacterial, anti-inflammatory activity as well as tissue regeneration capacity.”. Due to the previous studies that indicated that Opuntia ficus-indica extract could both inhibit the pathogenic species and  stimulate the wound healing process, we considered that it would be an effective compound in bacterial vaginitis infections, where Gardnerella vaginalis grows out of control and produces sialidase which disrupts the membrane integrity. 

Point 5. Why it is important to combine different concentrations of these compounds in the hydrogel formulations, and what is the overall goal in addressing vaginal infections and microbiota homeostasis?

Response 5: We considered that it is important to combine different concentrations of bioactive compounds because the final hydrogel formulation could not only have complementary contributions from these compounds, but be more than the sum of its components due to unexpected behaviors that arise from the compound interaction. Also, as we mentioned at lines 662-666, we added different cytocompatible and effective concentrations in the binary and ternary formulations in order to complement each other and improve the biological efficacy of the hydrogels for the treatment of vaginal infections as well as to prevent their recurrence and re-establish the homeostasis of vaginal microbiota.

Point 6. What is the significance of PX micelles, and how were they observed in the hydrogel structure?

Response 6: The PX micelles were observed by TEM microscopy and presented in Fig.2 and they represent the usual shape of PX copolymer at T>5°C observed by other researchers (ref 73-75 in the initial manuscript), in which the hydrophilic marginal PEG chains approach each other to close the micelle with hydrophobic PPG core. At lines 712-713 we mentioned the hypothesis that the variable diameter of micelles from 10 to 60 nm might be related to the encapsulation of biocompounds and we added now "especially hydrophobic compounds like Thyme essential oil" (lines 713-714). 

Point 7. What is the potential role of these hydrogels in supporting cell proliferation and promoting the growth of lactic acid bacteria?

Response 7: The increase in the prebiotic activity could be attributed to the polyphenol-rich content of plant extracts (Vitis vinifera seed extract, powder of Opuntia ficus-indica, which have direct prebiotic effects through the metabolization of polyphenols by probiotic microorganisms, especially those from the Lactobacillus genus. Previous studies demonstrated that phenolic compounds provide a favorable environment for the growth of lactic acid bacteria. In addition to this, other compounds with prebiotic activity, which were previously reported, are different types of carbohydrates like inulin, which is among the compounds accepted as a prebiotic by the International Scientific Association for Probiotics and Prebiotics (ISAP), as we mentioned now in the Introduction section (lines 96-99). Also, carbohydrates are known to be the main source of energy and without them different metabolic processes cannot take place properly, being thus indispensable for the metabolic activity of different cells.

Point 8. The author provide the outstanding points and highlights of this work in the conclusion section.

Response 8: Thank you for the recommendation, we have updated the conclusion section to be more descriptive of the main results.

Point 9. Typographical errors and superfluous spaces throughout the manuscript should be corrected.

Response 9: Thank you for the observation, we rechecked the manuscript for typographical errors.

Reviewer 3 Report

Comments and Suggestions for Authors

The manuscript is very complex, with a very current theme, with various and numerous methods of characterizing the obtained materials, and with interesting application.

But there are some things that should be changed:

- First of all, the title of the manuscript “Bioactive-loaded hydrogels based on bacterial nanocellulose, chitosan, and poloxamer for…” differs to some extent from the title presented on the website “Bioactive-loaded ternary hydrogels based on chitosan, nano-cellulose, and poloxamer for” …. Please correct this.

- The structure of the manuscript does not correspond to the Template, Materials and Methods must be before the Results

Abstract:

- line 23: “bioactive compounds” - it should be mentioned, in parentheses, which bioactive compounds were used.

- line 33: “L. reuteri and L. plantarum” - it should be mentioned what they are, for the uninitiated.

Keywords:

- too many keywords (10), especially since they are not all meaningful (e.g. lactic acid, citric acid). Authors should limit themselves to the most significant (maximum 6); also, replace “mucoadhesive hydrogel systems” with simple “mucoadhesion” or “mucoadhesiveness”.

Introduction:

- lines 49-51: Does it need mucoadhesive properties to increase also the pH stability and mechanical resistance? that's how it's understood. Or just to increase "transfer surface and contact time", but also require pH stability and mechanical resistance. The sentence needs to be revised to be understood well.

- line 86: “from many points of view” - it must be specified or justified why this area is extremely sensitive. Leaving the phrase as it is, it has no meaning.

-Line 96: “TEOS” - ???

-Line 102: “natural compounds/extracts” - The introduction does not mention at all the bioactive compounds added to the hydrogels, their specific properties and ultimately the purpose of using each individual component.

Results:

- lines 113-123: the description of the composition of the hydrogels should be placed in Materials and methods;

- line 177 - “characteristic citrate band”, line 184 – ”specificfor the aliphatic chains of fatty acids”; line 196 – “characteristic bands”; line 372 – “specific for aminoacids and proteins”; line 372 – “characteristic to COO− group”;

- when referring to a certain characteristic group or specific connection found in a compound, a reference is absolutely necessary! So, I recommend that a bibliographic reference be mentioned for each one.

- lines 235 and 238 – It should also be mentioned a reference to the two described models, Carreau and Hershel-Bulkley. Also line 418 – “Casson model” – reference?

Figure 12: What would be the explanation for the decrease in viability? which is the component in the composition of hydrogels that negatively affects viability?

Discussion:

- lines 575-611: This part should be shortened a little and added to the Introduction, because there the characteristics of each added component and the purpose of their use in hydrogels should be discussed at the beginning

- line 607: “we added different concentrations of…” - How were these concentrations, or these additions of the bioactive components, established? Following previous studies?

- lines 617-634: This paragraph about BNC is not necessary here. All these details are not related to the topics of the article, nor the chemical modification of BNC, nor nanocomposites based on BNC, etc. I recommend deleting this paragraph.

- line 706: “H1 and H2 should be kept at cold temperatures during long storage” - what would be the temperature required to keep them (the term "cold" is relative)? and also mentioned the storage term (min, max.).

- lines 809-810: “all hydrogels were active against E. coli” - It should be mentioned, however, that hydrogels are very little effective against E. coli, they are more than 90% effective only at 1.5x101, which is a very low bacterial density;

Materials and Methods:

- line 857: “4.2. Preparation of Hydrogels” - I recommend making a table from which it is clear what each hydrogel contains: H1, H2, H3... in the way it is described now, nothing is understood.

- line 878: “with and without mucin” - In the stage of obtaining the hydrogels, "mucin" is not mentioned. When is this added?

- line 894: “analyses (XRD) were performed on dried samples” - how were the samples dried?

- line 916: the equation for the adhesion energy “AE = F x d / S” should be numbered and separated from the text, to be more visible

- line 933: “The mucin binding efficiency was calculated…” - an equation should still be written, to see this calculation much better;

Conclusions:

- line 1025: “good stability” - how was this feature established? only visually? and for how long?

Author Response

The manuscript is very complex, with a very current theme, with various and numerous methods of characterizing the obtained materials, and with interesting application.

But there are some things that should be changed:

Author Response

Point 1: First of all, the title of the manuscript “Bioactive-loaded hydrogels based on bacterial nanocellulose, chitosan, and poloxamer for…” differs to some extent from the title presented on the website “Bioactive-loaded ternary hydrogels based on chitosan, nano-cellulose, and poloxamer for” …. Please correct this.

Response 1: Thank you for the observation. The final title will be without the word "ternary" in order to include also the two binary hydrogels H1 and H2.

Point 2: The structure of the manuscript does not correspond to the Template, Materials and Methods must be before the Results

Response 2:  We have checked again the Template of Pharmaceuticals Journal and apparently it did not change, being with the Results before the Materials and Methods. If requested by the Editors of Pharmaceuticals, we can update the manuscript accordingly.

Abstract:

Point 3: line 23: “bioactive compounds” - it should be mentioned, in parentheses, which bioactive compounds were used.

Response 3: Thank you. We mentioned them in parentheses.

Point 4: line 33: “L. reuteri and L. plantarum” - it should be mentioned what they are, for the uninitiated.

Response 4: Thank you. We added: „the probiotic strains Limosilactobacillus reuteri and Lactiplantibacillus plantarum subsp. plantarum”.

Point 5: Keywords: too many keywords (10), especially since they are not all meaningful (e.g. lactic acid, citric acid). Authors should limit themselves to the most significant (maximum 6); also, replace “mucoadhesive hydrogel systems” with simple “mucoadhesion” or “mucoadhesiveness”.

Response 5: According to the template provided on the website by Pharmaceuticals journal, we can list a maximum of 10 keywords and we tried to avoid any words that have already been used in the title, but at the same time to highlight the compounds that have been used in this study, which we consider an important aspect. We replaced the keyword “mucoadhesive hydrogel systems” with “mucoadhesion”.

Point 6: Introduction:

- lines 49-51: Does it need mucoadhesive properties to increase also the pH stability and mechanical resistance? that's how it's understood. Or just to increase "transfer surface and contact time", but also require pH stability and mechanical resistance. The sentence needs to be revised to be understood well.

Response 6: Thank you, we revised the phrase according to your suggestion.

Point 7: line 86: “from many points of view” - it must be specified or justified why this area is extremely sensitive. Leaving the phrase as it is, it has no meaning.

Response 7: We have provided additional justifications (lines 116-118).

Point 8: Line 96: “TEOS” - ???

Response 8: We added the explanation for the abbreviation TEOS on line 129: „tetraethyl orthosilicate”.

Point 9: Line 102: “natural compounds/extracts” - The introduction does not mention at all the bioactive compounds added to the hydrogels, their specific properties and ultimately the purpose of using each individual component.

Response 9: We have included additional Information according to your suggestion about reducing the Information from the Discussion section and inserting it in the Introduction section (lines 96-113).

Point 10: Results:

- lines 113-123: the description of the composition of the hydrogels should be placed in Materials and methods;

Point 10: We moved it in Section 4.2.

Point 11: - line 177 - “characteristic citrate band”, line 184 – ”specific for the aliphatic chains of fatty acids”; line 196 – “characteristic bands”; line 372 – “specific for aminoacids and proteins”; line 372 – “characteristic to COO− group”;

- when referring to a certain characteristic group or specific connection found in a compound, a reference is absolutely necessary! So, I recommend that a bibliographic reference be mentioned for each one.

Response 11: We have added relevant references for the specific FTIR bands.

Point 12: - lines 235 and 238 – It should also be mentioned a reference to the two described models, Carreau and Hershel-Bulkley. Also line 418 – “Casson model” – reference?

Response 12: We have added references for the rheological models.

Point 13: Figure 12: What would be the explanation for the decrease in viability? which is the component in the composition of hydrogels that negatively affects viability?

Response 13: Previous studies have shown the rich polyphenol content of Vitis vinifera seed extracts. However, although phenolic compounds are recognized for their high antioxidant activity, this property is often associated with the mechanisms by which these compounds induce cell death. Moreover, several studies indicated that ethanol can lead to the increase in reactive oxygen species (ROS) which further can lead to lipid peroxidation. Thus, by increasing the dose of hydrogels, the concentration of hydro-glycero-alcoholic Vitis vinifera seed extract increases, which could be a potential cause for the decrease in cell viability. Also, numerous studies have shown the dose-dependent antitumor activity of thyme essential oil. Moreover, the ratio of thyme essential oil compounds determines its bioactivity, for example it has been shown that a high content of monoterpenes such as thymol can lead to the disruption of membrane integrity, which could be also a potential cause for the decrease in cell viability by increasing the dose of hydrogels. This will be the subject of a further study, in which to optimize the composition of the hydrogels, by testing the optimal dose of individual components. We did not speculate too much on this issue, as we do not have any experimental proof in this sense.

Point 14: Discussion:

- lines 575-611: This part should be shortened a little and added to the Introduction, because there the characteristics of each added component and the purpose of their use in hydrogels should be discussed at the beginning

Response 14: Thank you. We shortened it a little and moved it in the Introduction section.

Point 15: - line 607: “we added different concentrations of…” - How were these concentrations, or these additions of the bioactive components, established? Following previous studies?

Response 15: We chose to add in the hydrogel bases concentrations of bioactive compounds inspired from the literature and we tested different doses of hydrogels to select the optimal dose that could have the potential to rebalance the vaginal microbiota.

Point 16: - lines 617-634: This paragraph about BNC is not necessary here. All these details are not related to the topics of the article, nor the chemical modification of BNC, nor nanocomposites based on BNC, etc. I recommend deleting this paragraph.

Response 16: In this paragraph we started to describe the main biopolymers used in our hydrogels, therefore a part of this paragraph regarding BNC was kept and the less relevant information was deleted, as suggested.

Point 17: - line 706: “H1 and H2 should be kept at cold temperatures during long storage” - what would be the temperature required to keep them (the term "cold" is relative)? and also mentioned the storage term (min, max.).

Response 17: Thank you for suggestion, we have changed "cold" with "refrigerating" temperatures (3-5°C). We also added "Although not particularly analyzed, the first batches dating 7 months ago still show a good homogeneity and consistency in refrigerating storage conditions, but this aspect will be further investigated."

Point 18: - lines 809-810: “all hydrogels were active against E. coli” - It should be mentioned, however, that hydrogels are very little effective against E. coli, they are more than 90% effective only at 1.5x101, which is a very low bacterial density;

Response 18: We added more information (lines 870-877).

Point 19:  Materials and Methods:

- line 857: “4.2. Preparation of Hydrogels” - I recommend making a table from which it is clear what each hydrogel contains: H1, H2, H3... in the way it is described now, nothing is understood.

Response 19: We added in this section the Table 3 which contains the composition of each hydrogel and hydrogel base to make it easier to follow the preparation steps described in section 4.2.

Point 20: - line 878: “with and without mucin” - In the stage of obtaining the hydrogels, "mucin" is not mentioned. When is this added?

Response 20: We added more information.

Point 21: - line 894: “analyses (XRD) were performed on dried samples” - how were the samples dried?

Response 21: We added more information (line 981).

Point 22: - line 916: the equation for the adhesion energy “AE = F x d / S” should be numbered and separated from the text, to be more visible

Response 22: We have separated and numbered the equation as suggested.

Point 23: - line 933: “The mucin binding efficiency was calculated…” - an equation should still be written, to see this calculation much better;

Response 23: We added the equation.

Point 24: Conclusions: - line 1025: “good stability” - how was this feature established? only visually? and for how long?

Response 24: We have added after good stability: "under dynamic shearing conditions and visually under 7-months refrigerating storage conditions" (lines 1120-1121). The hydrogel stability under dynamic conditions was investigated by rheology with hysteresis loops, the similitude between the UP-ramp and reverse ramp suggesting the stability in shearing conditions, as discussed in the sub-section 2.1.3. Rheological properties of the hydrogels. We also added the stability in refrigerating storage conditions that we mentioned previously. We hope the conclusions are clearer now and we thank you again very much for all your useful recommendations.

Reviewer 4 Report

Comments and Suggestions for Authors

I have reviewed manuscript entitled, "Bioactive-loaded ternary hydrogels based on chitosan, nano-cellulose, and poloxamer for rebalancing vaginal microbiota". Manuscript is written well and presented in general. Its a significant work conducted to develop ternary hydrogels for balancing of vaginal microbiota. I have following concerns those must be addressed for further processing;

1. Authors have developed a polymeric hydrogel system for delivery of bioactive agents in order to rebalance vaginal microbiota. They have successfully developed biocompatible systems with desired utility.  

2. The topic is original and suits the Pharmaceutical field as it involves development of ternary hydrogels.   

3. It involves the introduction of a new delivery approach instead of conventional delivery of drugs with an aid of some support.  

4. Development process of ternary hydrogels can be improved further.  

5. Add significant results in abstract section.  

6. In introduction section provide advantages of ternary hydrogels over hydrogels?  

7. Conclusions are ok but can be improved by adding limitations of work in it.  

8. References can be updated from last  year's literature.  

9. Figures 12 to 16 please indicate what is meant for a, b, c, ab within the figures indicated?  

10. Thoroughly revise entire manuscript.

Author Response

Point 1: I have reviewed manuscript entitled, "Bioactive-loaded ternary hydrogels based on chitosan, nano-cellulose, and poloxamer for rebalancing vaginal microbiota". Manuscript is written well and presented in general. Its a significant work conducted to develop ternary hydrogels for balancing of vaginal microbiota. I have following concerns those must be addressed for further processing;

Authors have developed a polymeric hydrogel system for delivery of bioactive agents in order to rebalance vaginal microbiota. They have successfully developed biocompatible systems with desired utility.  

The topic is original and suits the Pharmaceutical field as it involves development of ternary hydrogels.   

It involves the introduction of a new delivery approach instead of conventional delivery of drugs with an aid of some support.  

Development process of ternary hydrogels can be improved further.  

Point 1. Add significant results in abstract section.  

Response 1: Thank you very much for your appreciations on our paper. We have revised the abstract with additional relevant results.

Point 2. In introduction section provide advantages of ternary hydrogels over hydrogels?  

Response 2: Thank you for your suggestion, we have added at the end of Introduction a paragraph regarding the possible advantages of ternary hydrogels over binary hydrogels (lines 130-133).

Point 3. Conclusions are ok but can be improved by adding limitations of work in it.  

Response 3: We have revised the conclusions and added limitations at the end.

Point 4. References can be updated from last year's literature.  

Response 4: Thank you, we have updated the literature.

Point 5. Figures 12 to 16 please indicate what is meant for a, b, c, ab within the figures indicated?  

Response 5: We mentioned in the figure legend that “different letters show statistically significant differences between the samples”.

Point 6. Thoroughly revise entire manuscript.

Response 6: Thank you. We checked.